# Recurrent evolution of ligand-binding domain multiplicity fine-tunes TGFβ signaling in vertebrates

Jerome Jatzlau [1,8] ✉, Michael Trumpp [1,2,8], Julia Kühlwein [1], Leon Obendorf [1,2], Yao Le [3], Heiner Kuhl [4,5], Marco Preussner [1], Paul-Lennard Mendez[1], Hendrik Burkert[1], Wiktor Burdzinski [1,6], Stefan Mundlos [7], Christoph Winkler [3], Matthias Stöck [4] & Petra Knaus [1,6] ✉

From sponges to mammals, TGFβ signalling is a central regulator of body plan, cell fate and tissue homeostasis, with receptor architecture highly conserved across metazoans. Here we identify unexpected evolutionary divergence within a key receptor structure: 12 independent ligand-binding domain (LBD) duplications across three receptor classes (ACVR1, BMPR2 and TGFBR2) in diverse vertebrate lineages, including fish, amphibians, birds, and mammals. These duplications reveal previously unrecognized structural and functional plasticity arising from domain-level innovation, including in established model organisms such as zebrafish, African clawed frog and chicken. Recently diverged lineages conserve the membrane-distal LBD and ligand-interacting residues, correlating with enhanced ligand binding, whereas more ancient lineages exhibit elevated evolutionary rates of the membrane-distal LBD associated with inhibitory function. Our findings reveal LBD multimerization as a recurring, lineage-independent mechanism that diversifies and fine-tunes TGFβ signalling, adding a regulatory dimension to one of the best-examined conserved and essential pathways in metazoan biology.

Signaling through ligand-binding receptors has been a cornerstone of multicellular life since the early evolution of metazoans[1]. Multiplication of single exon-coded domains shaped the extracellular architectures of many growth factor receptors. Some, such as EGF-, FGF-, and VEGF receptors, contain repeated ligand-binding domains (LBDs), yet the number and arrangement of these repeats have remained conserved from early metazoans to vertebrates[2–5]. By contrast, other key ligand-binding receptors such as those of the TGFβ receptor family have been supposed to contain only a single LBD,

conserved from invertebrates to mammals. Remarkably, we found an exception in this otherwise highly conserved pathway: in medaka (*Oryzias latipes*), a single exon encoding the ligand-binding domain in *ACVR1* underwent triplication, producing a receptor with three tandem LBDs[6].

The TGFβ-receptor family, comprising TGFβ and bone morphogenetic protein (BMP) receptors, represents a highly conserved signaling system, essential for regulating developmental and homeostatic processes across all vertebrates[7]. Their roles span from orchestrating

[1]Freie Universität Berlin, Institute of Chemistry and Biochemistry - Biochemistry, Berlin, Germany. [2]International Max Planck Research, School for Biology and Computation, Berlin, Germany. [3]Department of Biological Sciences and Centre for Bioimaging Sciences, University of Singapore, Singapore, Singapore. [4]Department of Fish Biology, Fisheries and Aquaculture, Leibniz Institute of Freshwater Ecology and Inland Fisheries, Berlin, Germany. [5]Ecotoxicological Laboratory, German Environment Agency (UBA), Berlin, Germany. [6]Berlin-Brandenburg School for Regenerative Therapies (BSRT), Berlin, Germany. [7]Max Planck Institute for Molecular Genetics, Berlin, Germany. [8]These authors contributed equally: Jerome Jatzlau, Michael Trumpp. ✉e-mail: jerome.jatzlau@fu-berlin.de; petra.knaus@fu-berlin.de

early embryonic tissue formation to fine-tuning cellular identities across diverse cell lineages[8,9] These receptors share a characteristic organization, an extracellular LBD followed by a transmembrane (TM) and an intracellular kinase domain (KD)[10,11], that is preserved from invertebrates to vertebrates.

Following two rounds of whole-genome duplication (WGD) events, the number of TGFβ receptor genes expanded from five in invertebrates[12] to thirteen in vertebrates[13], of which twelve are retained in mammals, including five type II and seven type I receptors[14]. This expanded receptor repertoire co-evolved with more than thirty TGFβ ligand family members, forming highly specific ligand–receptor pairs[14]. Upon ligand binding to their high-affinity receptors, a tetrameric complex of two type I and two type II receptors assembles, enabling transphosphorylation and activation of the type I receptor kinase by a constitutively active type II kinase. Activated type I receptors phosphorylate intracellular mediators, most commonly SMAD transcription factors, which regulate the expression of TGFβ/BMP target genes in a cell type- and context-dependent manner to control key developmental decisions[8].

In this study we investigated whether LBD multiplications within the TGFβ receptor family are more widespread as previously recognized and how such structural innovations affect receptor signaling. Through a comprehensive analysis across vertebrate genomes, we identified 12 independent cases of convergent LBD multiplication in

three TGFβ receptor genes, spanning a broad phylogenetic range from fish to mammals. These events were detected in multiple species, including elephant-nose fish, carps, clawed frogs, chicken and horses, highlighting that LBD multimerization is a recurrent evolutionary phenomenon across vertebrate lineages. Our findings reveal that these can substantially modulate receptor signaling properties. Thus, LBD multiplication represents a previously unknown mode of molecular innovation within the highly conserved TGFβ signaling system, adding a new layer of complexity and regulatory potential to this essential developmental pathway.

## Results

### WGD-independent LBD expansion in TGFβ receptors

Previously, we identified and verified that the medaka *ACVR1* paralogs, derived from teleost-specific whole genome duplication, differ in their number of LBDs[6]. Whereas the medaka *ACVR1* carries three LBDs, the *ACVR1L* gene encodes for a BMP type I receptor with only 1 LBD. In contrast, zebrafish (*Danio rerio*) has lost the *ACVR1* gene and retained only *ACVR1L* with 1 LBD. To trace the origin of the ACVR1 LBD triplication, we compared available genome and transcriptome data across various ray-finned fish (Actinopterygii) lineages, supplementing missing data with RNA sequencing of heart tissues from selected representatives of individual families (Fig. 1). We found that an initial duplication of the *ACVR1* LBD occurred in the early ancestors of

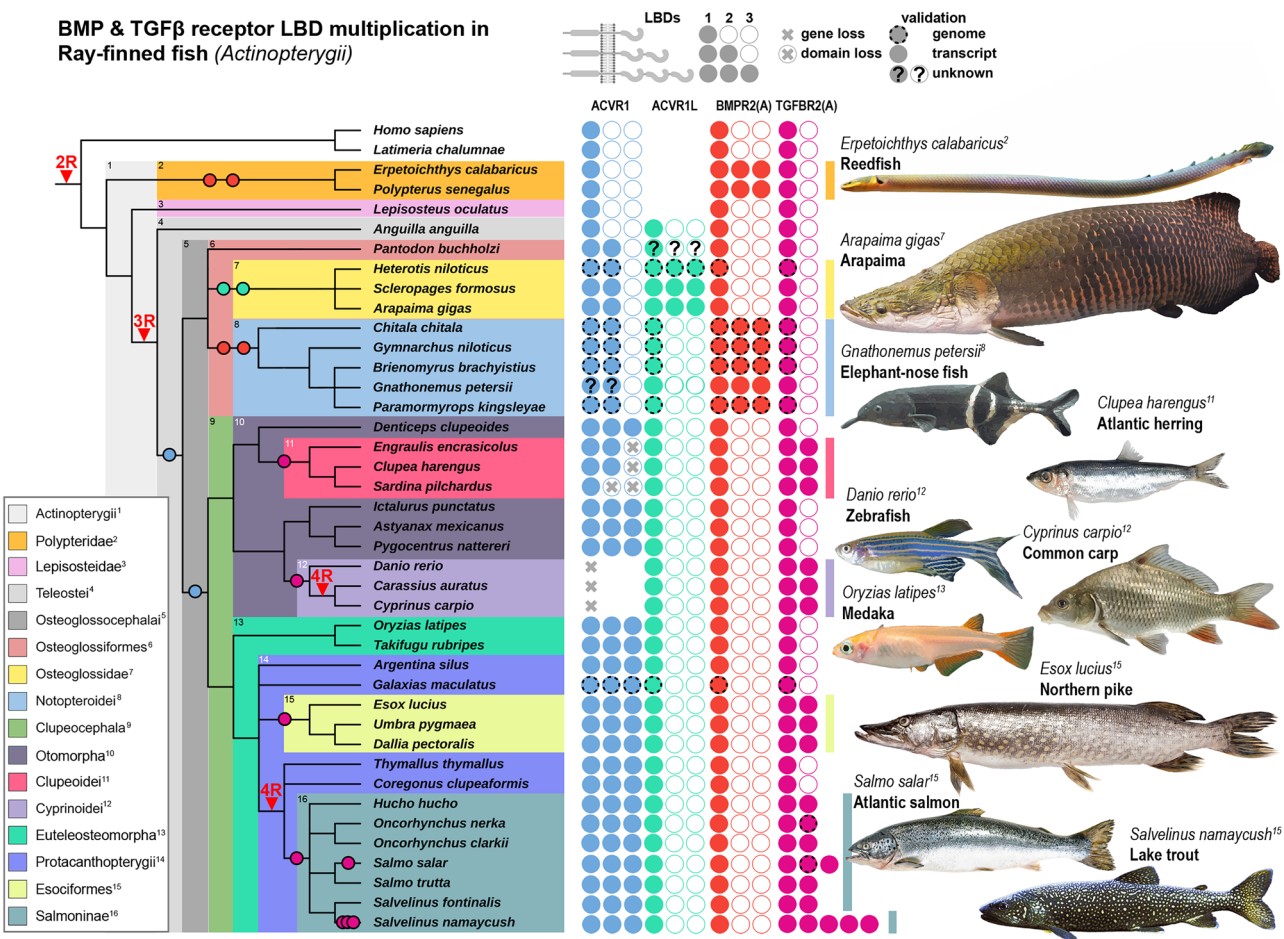

**Fig. 1 | Convergent evolution of TGFβ & BMP-Receptor LBD multiplication in ray-finned fish.** Evolutionary tree displaying the relationship between different ray-finned fish clades. Ligand Binding Domain (LBD) duplication events (circles) occurred at least 5 times in four BMP and TGFβ-Receptor genes homologs (ACVR1, ACVR1L, BMPR2 and TGFBR2). Additionally, LDB triplication events have occurred 4 times subsequently or in parallel (ACVR1, ACVR1L and BMPR2). Vertebrate-specific (2 R), teleost-specific (3 R) and carp-specific (4 R) genome duplication events are indicated. Number of filled circles represent 1 to 3 LBDs. Transcripts are either validated by RNA-seq or predicted genes from available genomes assemblies (dashed-line). Schematics created in BioRender. Trumpp, M. (https://BioRender.com/8p64dhh).

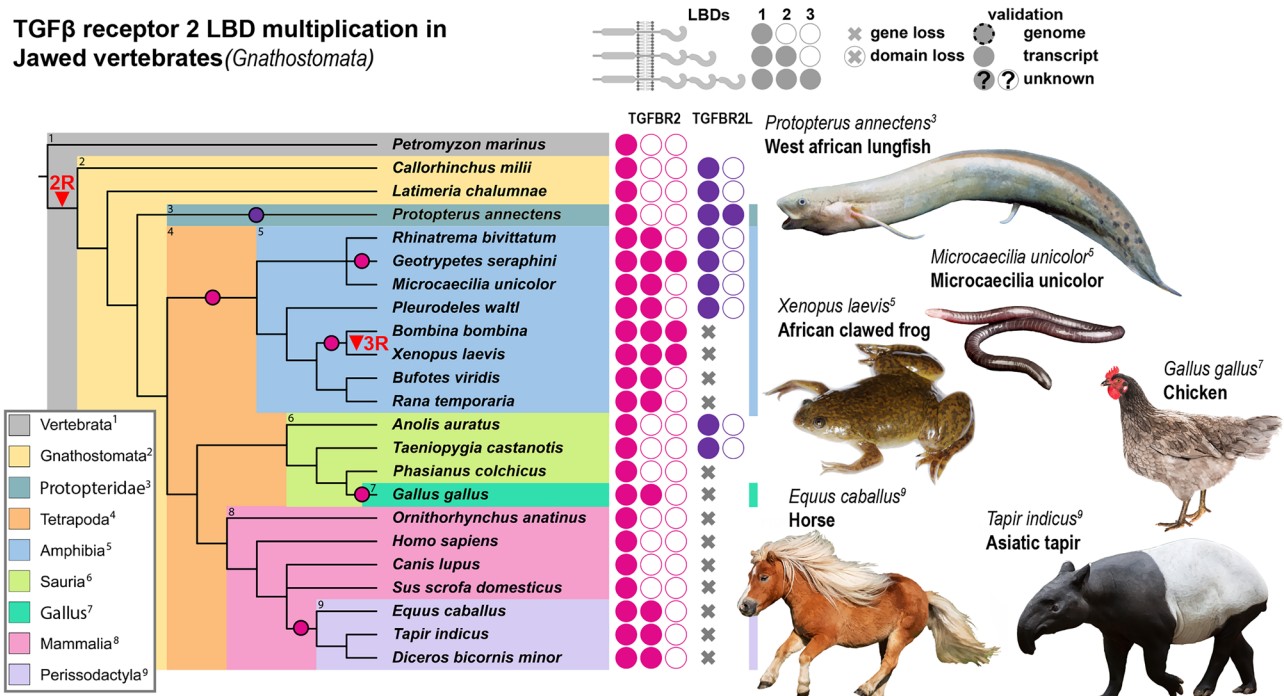

**Fig. 2 | Convergent LBD multiplication in TGFBR2 orthologs of jawed vertebrates.** Evolutionary tree displaying the relationship of jawed vertebrates. Ligand Binding Domain (LBD) duplication events (circles) occurred at least 4 times in the TGFβ-Receptor type 2 genes paralogs *TGFBR2* and *TGFBR2L*. Additionally, TGFBR2 LDB triplication events have occurred 2 times independently in different amphibian clades. Vertebrate-specific ancestral whole-genome duplication (2 R) is indicated and gave rise to the paralogs *TGFBR2* and *TGFBR2L*. *TGFBR2L* has been lost in all mammals, some bird and amphibian species. Number of filled circles represent 1 to 3 LBDs. Transcripts are all validated by RNA-seq. Schematics created in BioRender. Trumpp, M. (https://BioRender.com/8p64dhh).

Osteoglossocephalai, followed by a secondary triplication event common to all Clupeocephala. Interestingly, individual LBDs were subsequently lost within herrings, anchovies and shads (Clupeoidei), and the entire *ACVR1* gene was lost in all carp-like fishes (Cyprinoidei). We next asked whether LBD multiplication is specific to the *ACVR1* locus, or if it is a common feature across all teleost genomes following evolutionary adaptation after the teleost whole genome duplication event[15]. Surprisingly, we found an LBD triplication event in the *ACVR1* paralog *ACVR1L*, which is unique to the bonytongues (Osteoglossidae). We uncovered six additional LBD multiplication events in the type II receptor genes *BMPR2* and *TGFBR2*, indicating a broader pattern beyond the *ACVR1* locus. Notably, the *BMPR2* gene underwent LBD triplication twice, once within the bichirs (Polypteridae) and once the featherbacks and knifefishes (Notopteroidei). These events appear to be independent of *BMPR2* paralogy, as Polypteridae diverged before the teleost whole genome duplication. The *TGFBR2* locus underwent four independent LBD multiplication events in Clupeoidei, Cyprinoidei, pikes and mudminnows (Esociformes), and salmonids (Salmoninae). While the carp-specific *TGFBR2* LBD duplication preceded the carp-specific whole genome duplication (WGD) (Supplementary Fig. 1a), the *TGFBR2* LBD multimerization in Salmoninae occurred after the salmon-specific WGD. Finally, we extended our analysis to all jawed vertebrates (Gnathostomata) (Fig. 2) to test whether TGFβ-superfamily receptor LBD multiplication depends on preceding WGD. We identified four additional TGFBR2 LBD multiplication events: one in the *TGFBR2* paralog *TGFBR2L* within the african lungfish (Protopteridae), and three in *TGFBR2* within amphibians (Amphibia), chicken (Gallus)[16], and odd-toed hoofed mammals (Perissodactyla). Collectively, we identified 12 convergent LBD multiplication events across three TGFβ-family receptor genes, *ACVR1/ACVR1L*, *BMPR2*, and *TGFBR2/TGFBR2L*, in species that had undergone WGD and retained a backup paralog (e.g., *ACVR1* in all teleosts, or *TGFBR2* in Salmoninae). However, we also observed LBD multiplication in species with only a single gene copy, such as BMPR2 in Polypteridae and *TGFBR2* in Perissodactyla, suggesting that WGD is not a prerequisite for LBD multimerization. Interestingly, type I receptor LBDs are encoded by a single exon, whereas type II receptor LBDs are split across two exons, highlighting that, in the case of *BMPR2* and *TGFBR2*, duplication of an entire exon pair was required (Supplementary Fig. 2a). Since the *TGFBR2* locus exhibited the highest number of LBD duplication events, we next investigated whether repetitive elements flanking exons 2 and 3 of *TGFBR2* are conserved across species, potentially facilitating repeated independent duplications. Whereas in the zebrafish *Tgfbr2a* locus, retroelement derived GT-rich sequences are framing the duplicated exon pair, these or other repetitive sequences are absent in the chicken (*Gallus gallus*) and horse (*Equus caballus*) *TGFBR2* loci at duplication sites (Supplementary Fig. 2b). The duplication sizes range widely, from ≥2.1 kb in zebrafish to ≥18.7 kb in horse, indicating that these exon-pair duplications arose through separate genetic events rather than from a shared ancestral duplication (Supplementary Fig. 2b).

### Membrane-proximal LBD drives gnBmpr2a signaling

To assess the molecular function of convergently multimerized LBDs, we first analysed the sequence conservation among 3 LBDs within individual BMPR2 proteins of the Notopteroidei and Polypteridae lineages (Fig. 3a, and Supplementary Fig. 3). In both groups, the LBDs exhibited intermediate levels of sequence identity (50-70%) relative to the membrane-proximal LBD (LBD^in), suggesting partial functional divergence within the repeats (Fig. 3a). BMPR2 functions as a type II receptor for a wide range of BMP and Activin ligands[17,18]. Whereas BMP binding to BMPR2-containing receptor complexes requires the presence of a high affinity type I receptor, BMPR2 can directly bind Activin A through a specific set of hydrogen bonds and hydrophobic interactions primarily involving residues located between fingers 2 and 3 (Fig. 3b, c)[19].

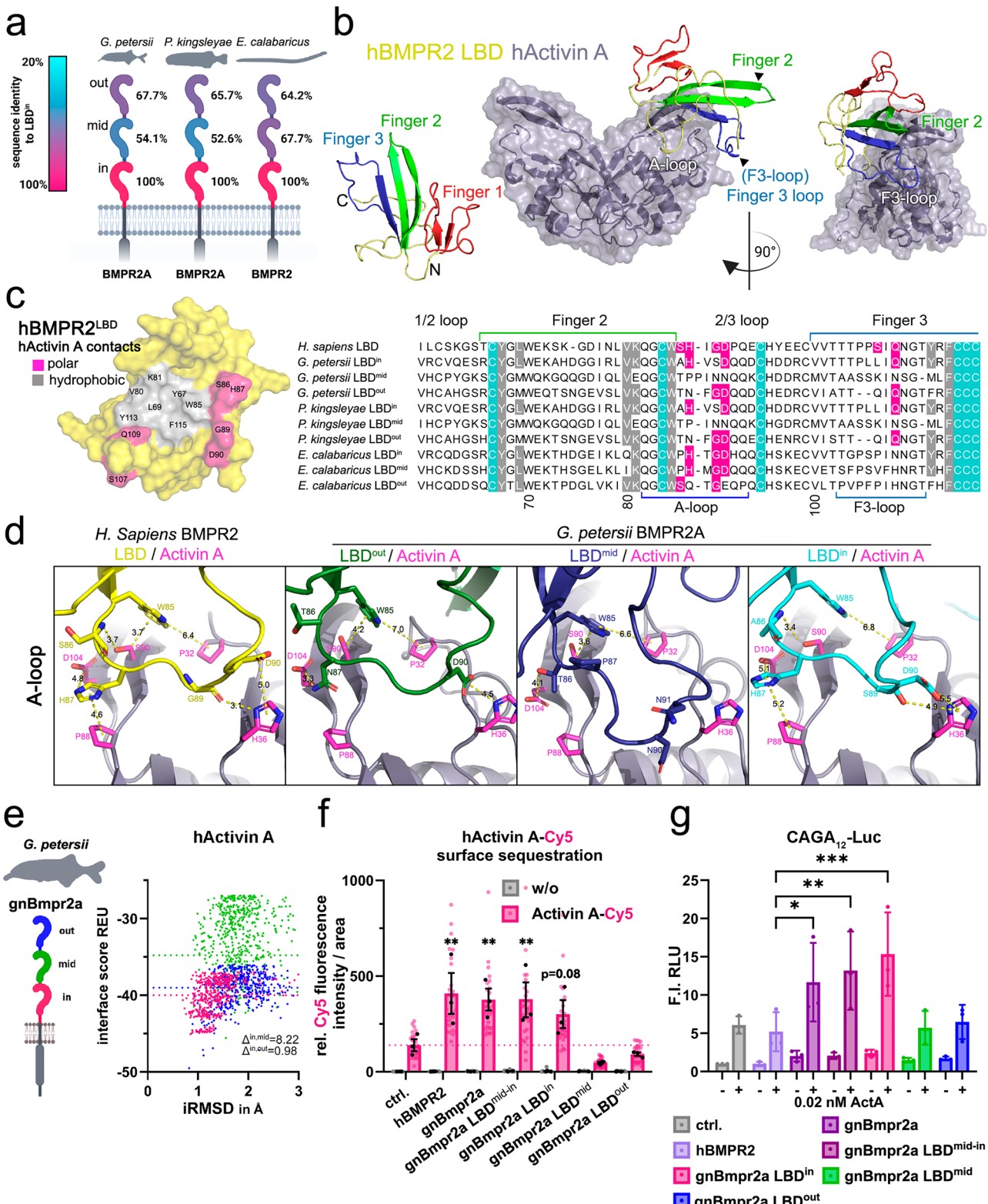

To determine the conservation of these critical interaction sites, we compared the BMPR2 LBD residues involved in ligand binding across the 3 domains. In both lineages, the inner LBD showed the highest degree of sequence conservation with human BMPR2, particularly at residues forming the Activin-specific binding interface (Fig. 3c). Interestingly, the inner LBDs, of Notopteroidei species retained 85% of the key residues involved in A-loop contacts in finger 2, while Polypteridae retained 69% (Fig. 3c, d). Consistent with this, members of the Notopteroidei also exhibited minimal loss of conserved interaction-mediating residues on finger 3, consistent with a maintained or partially preserved capacity of Activin binding in LBD$^{in}$ (Fig. 3d, and Supplementary Fig. 4a).

To explore the functional impact of LBD triplication in BMPR2 receptors from Notopteroidei, we performed computational protein-protein docking using the Rosetta Commons suite[20]. Docking simulations were conducted using the triplicated LBD of BMPR2a from

**Fig. 3 | Inner LBD conveys signaling competence of LBD multimerized BMPR2 ortholog. a** Scheme of BMPR2 LBD sequence identity comparison relative to respective inner LBD within *G. petersii*, *G. niloticus* and *E. calabaricus*. **b** Structure of hActivin A with the LBD interface of hBMPR2 obtained via Rosetta docking starting from an AlphaFold2-multimer prediction. Interaction is facilitated by finger motif 3 (F3-loop) and A-loop of receptor LBD. **c** Molecular surface of hBMPR2 LBD showing the hActivin A interface, residues forming polar or hydrophobic contacts are colored in pink and gray, respectively (**left**). Sequence alignment of BMPR2 LBDs of *H. sapiens*, *G. petersii*, *G. niloticus* and *E. calabaricus* LBDs with hActivin A contact residues and critical backbone cysteines (teal) highlighted; interaction sites F3-loop and A-loop are indicated with colored lines (**right**). **d** Cartoon and stick representation of respective animal receptor LBD interaction with hActivin A (A-loop) based on AlphaFold2-multimer models PyMOL was used for image representation. **e, left** Illustration of *G. petersii* Bmpr2a 3 LBD receptor domain structure as reference, highlighting the inner (pink), middle (green) and outer (blue) ligand binding domain. **e, right** In silico interface analysis via Rosetta docking of hActivin A interactions with single gnBmpr2a domains, depicted as interface score (REU) in relation to interface root mean square deviation (iRMSD) in angstroms. Differences of the mean REU of out and mid LBD are calculated against inner LBD. **f** Activin A-Cy5 surface binding to Halo-hBMPR2 and Halo-gnBmpr2a variants is represented

as relative fluorescence intensity per area. **f** Activin A−Cy5 surface binding to Halo-hBMPR2 and Halo-gnBmpr2a variants, shown as relative fluorescence intensity per area (± Activin A−Cy5). Data are mean ± SEM. $n = 3$ independent biological replicates (independent experiments; up to 10 cells per experiment averaged). Two-way ANOVA with Dunnett's multiple comparisons test (two-sided; vs ctrl.) was used. Exact $P$ values: hBMPR2 vs ctrl., $p = 0.0015$; gBmpr2a vs ctrl., $p = 0.0052$; gBmpr2a LBD$^{mid\text{-}in}$ vs ctrl., $p = 0.0056$; gBmpr2a LBD$^{in}$ vs ctrl., $p = 0.0825$. Error bars represent SEM; black dots indicate independent biological replicates, and grey and pink dots represent individual cells. The dotted line denotes ctrl. Activin A−Cy5 levels. **g** pSMAD2/3-sensitive CAGA$_{12}$ luciferase reporter activity of Halo-gnBmpr2a receptor variants in the presence or absence of Activin A (0.02 nM), shown as relative light units (RLU) and expressed as fold induction (F.I.) relative to control (− Activin A). Data are mean ± SD. $n = 3$ independent biological replicates (independent experiments). Two-way ANOVA followed by Dunnett's multiple comparisons test (two-sided; vs hBMPR2A) was used. Exact $P$ values: gBMPR2A vs hBMPR2A, $p = 0.0304$; gBMPR2A LBD$^{mid\text{-}in}$ vs hBMPR2A, $p = 0.0057$; gBMPR2A LBD$^{in}$ vs hBMPR2A, $p = 0.0004$. Error bars represent SD. Source Data are provided as a Source Data file. **a, e** Schematics created in BioRender. Trumpp, M. (https://BioRender.com/8p64dhh).

elephant-nose fish (*Gnathonemus petersii;* member of the Notopteroidei), and human Activin A as the ligand (Fig. 3e, and Supplementary Fig. 4b). Additionally, lineage-specific docking analyses were performed using the appropriate Activin A orthologs for both Notopteroidei and Polypteridae species (Supplementary Fig. 4e–g). Consistent with the LBD conservation analysis (Fig. 3a–c), the inner LBD showed the lowest predicted interface energy, indicating the highest ligand-binding affinity (Fig. 3e). To experimentally test the functionality of the individual LBDs, we generated expression constructs of elephant-nose fish BMPR2a, including full-length receptors, deletion mutants containing only one or two LBD (LBD$^{mid\text{-}in}$). Binding of fluorescently-labelled human Activin A-Cy5 was assed using ligand surface binding assay (LSBA, Fig. 3f; and Supplementary Figs. 4c, 5)[21], and downstream signaling was quantified via a CAGA$_{12}$-driven dual luciferase assay, which reports SMAD2/3 activity (Fig. 3g; and Supplementary Fig. 4d).

In agreement with the docking predictions, only the inner, membrane-proximal LBD bound Activin A with high affinity. Neither the outer, nor the middle LBD alone showed detectable binding, while LBD$^{in}$ and LBD$^{mid\text{-}in}$ exhibited comparable binding to the full-length receptor (Fig. 3f). This binding pattern was mirrored in the functional signaling output: constructs containing only the outer or middle LBD showed minimal SMAD2/3 activation, whereas LBD$^{in}$ displayed the highest signaling competence (Fig. 3g). Notably, deletion of the outer and middle LBDs enhanced signaling efficiency, indicating that these LBDs are functional but act repressively, whereas only the membrane-proximal inner LBD contributes positively to Activin A responsiveness in the LBD triplicated receptor (Fig. 3g). Interestingly, the BMPR2 ortholog from elephant fish enhanced both Activin A binding and SMAD2/3 signaling, whereas the human BMPR2 control increased ligand binding but failed to efficiently promote SMAD2/3 activation. This functional divergence suggests that the elephant fish BMPR2 exhibits ACVR2-like receptor behavior[22].

**Divergent evolution of TGFBR2 LBD multimers tunes signaling**
Next, we examined the TGFBR2 gene, which independently acquired LBD multiplications in seven distinct lineages, followed by additional expansions in salmonids and amphibians, as seen in atlantic salmon (*Salmo salar;* 3 LBDs) and lake trout (*Salvelinus namaycush;* 5 LBDs), gaboon caecilian (*Geotrypetes seraphini;* 3 LBDs), and african clawed frog (*Xenopus laevis;* 3 LBDs) (Figs. 1, 2). To predict the molecular function of the multimerized LBD receptors, we first compared the LBD sequence homology within one protein (Fig. 4a, and Supplementary Fig. 6). We observed that the degree of conservation varied among the seven families. While the carp (*Cyprinus carpio*) and clawed

frog TGFBR2 LBDs showed the lowest conservation (25–50%) relative to the membrane-proximal LBD (LBD$^{in}$), pike (*Esox lucius*) and herring (*Clupea harengus*) exhibited intermediate conservation (50–90%), and the salmons, chicken, and horse displayed high conservation levels (90–100%). (Fig. 4a). TGFBR2 is known for its high affinity binding to TGFβ-ligands via polar and hydrophobic interactions of residues mainly residing in finger 1 of the LBD (Fig. 4b, c)[23]. When comparing the amino acids, which form the interaction epitope for human TGFβ1 within TGFBR2, these are highly conserved in the LBD$^{in}$ domain of all tested species, whereas the middle or outer LBD domains show a lower degree of conservation (Fig. 4c, and Supplementary Fig. 7). Interestingly, consistent with its overall low degree of conservation, the zebrafish Tgfbr2a LBD$^{out}$ shows no conserved TGFβ1-binding amino acid residues (Fig. 4c, d). In contrast, the chicken LBDs exhibit 98% overall similarity and share an identical TGFβ1-binding epitope (Fig. 4c, d). Lastly, the horse and tapir LBD$^{out}$ show a high level of overall conservation (93.1%, 96%), with three or one amino acid substitutions within the TGFβ1-binding epitope, respectively (Fig. 4c, d; and Supplementary Fig. 7). In contrast, to the varying degree of conservation on the side of the LBD interface, all TGFβ1 ligand orthologs show a complete conservation of amino acids forming the receptor interacting interface (Supplementary Fig. 7). Finally, to evaluate whether the extent of duplicated LBD sequence divergence corresponds to the evolutionary time separating each lineage, we calculated molecular clocks using orthologous LBD domains. The resulting lineage-specific clocks indicated that the LBD duplications in horse, chicken, and salmon evolved at rates consistent with their respective orthologous LBDs, indicating that these domains remain highly conserved under a clock-like model of evolution. However, the zebrafish LBD duplication strongly deviated from this pattern, exhibiting an exceptionally high substitution rate, far exceeding that of the clade-specific (otomorpha) LBD molecular clock, indicative of accelerated evolution (Supplementary Fig. 8).

To functionally predict the effect of TGFBR2 LBD duplication on ligand binding capabilities we again performed protein-protein docking studies (Fig. 5a) using human TGFβ1 and the respective animal ortholog TGFβ1, which exhibit a high degree of conservation (Fig. 5, and Supplementary Figs. 9,10,11). Furthermore, we cloned zebrafish, carp, clawed frog, chicken and horse TGFBR2 expression constructs, and generated single LBD mutants, to separately test the functionality of individual LBDs. Using fluorescently labeled human TGFβ1-SiRd12, we performed LSBA[21] to verify the predicted docking data (Fig. 5b) and tested the downstream signaling response using the SMAD2/3-sensitive CAGA$_{12}$-dual luciferase assay (Fig. 5c). In line with the low degree of

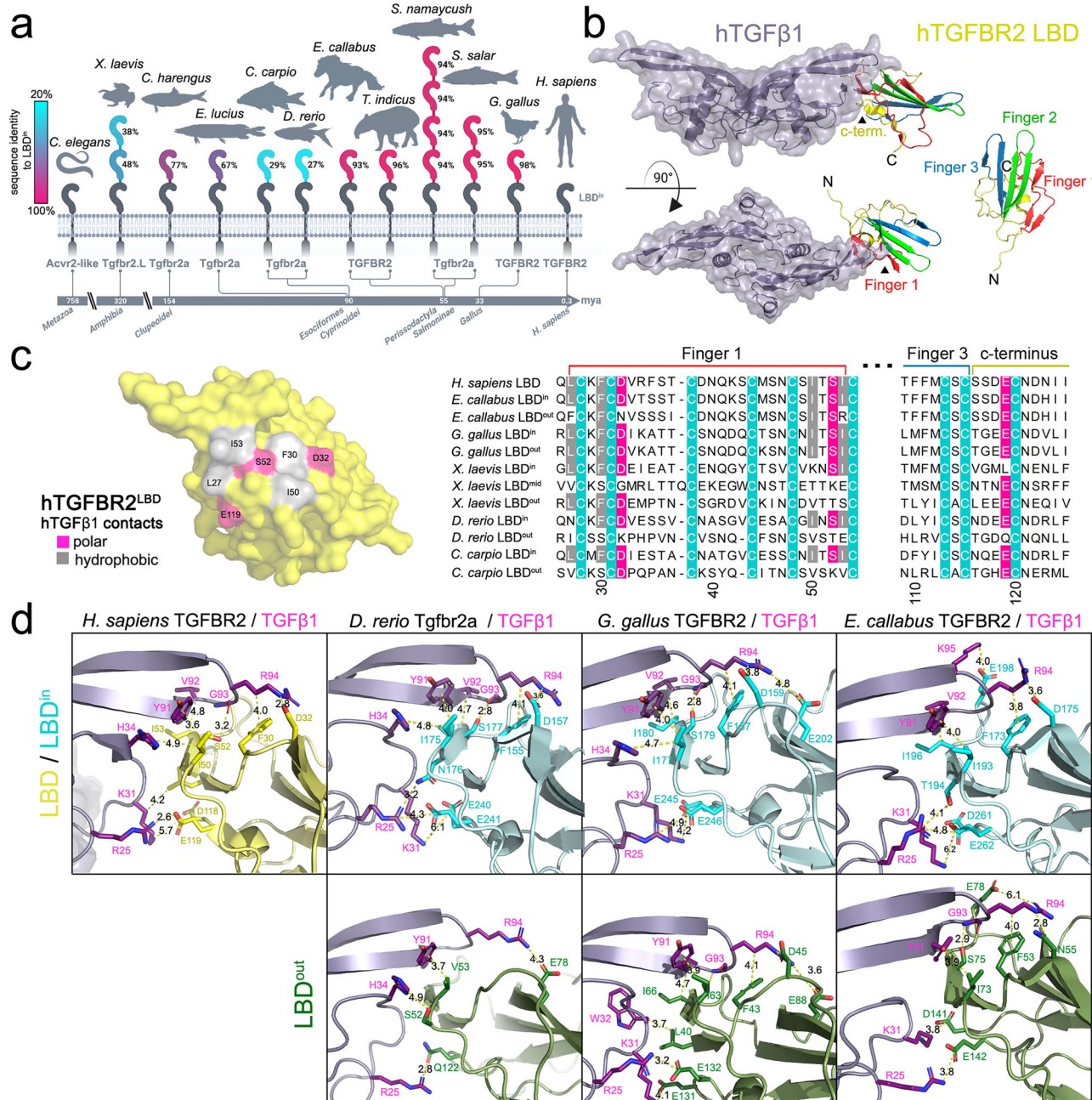

**Fig. 4 | Multimerized LBDs of TGFBR2 orthologs display varying degree of TGFβ1-interface conservation. a** Scheme of TGFBR2 LBD sequence identity comparison relative to respective inner LBD within each depicted animal. **b** Structure overview of hTGFβ1 with the LBD interface of hTGFBR2 obtained via Rosetta docking starting from an AlphaFold2-multimer prediction. Interaction is facilitated by c-terminus, finger motif 1 and 3 of receptor LBD. **c** Molecular surface of hTGFBR2 LBD showing the hTGFβ1 interface residues forming polar or hydrophobic contacts are colored in pink and gray, respectively (**left**). Sequence alignment of TGFBR2 LBDs of *Homo sapiens, Equus caballus, Gallus gallus, Xenopus laevis, Danio rerio, Cyprinus carpio* with TGFβ1 contact residues and critical backbone cysteines (teal) highlighted; interaction sites Finger 1, 3 and c-terminus are indicated with colored lines above (**right**). **d** Cartoon and stick representation of respective animal receptor LBD interaction with hTGFβ1 (c-terminus, finger motif 1 and 3) based on AlphaFold2-multimer models docked to hTGFβ1 using the lowest energy structure after performing a Rosetta docking protocol. PyMOL was used for image representation. **a** Schematics created in BioRender. Trumpp, M. (https://BioRender.com/8p64dhh).

LBD conservation in both zebrafish and carp Tgfbr2a, the LBD[out] exhibits almost no TGFβ1 binding, whereas the LBD[in] domain is capable of TGFβ1 binding and transmitting signals (Fig. 5d–f, Supplementary Figs. 9a–c, 12, 13). Strikingly, only in the zebrafish (Figure 5e, Supplementary Fig. 12), and reduced SMAD2/3 activity as seen in the reporter gene assay (Fig. 5f).

Similarly, in horse and frog TGFBR2 orthologs, the membrane-proximal domain LBD[in] shows the highest predicted and experimentally

confirmed binding to TGFβ1, while the outer domain (LBD[out]) exhibits reduced or no binding and lacks signaling competence (Fig. 5g–i, Supplementary Figs. 9d–f, 14, 15). Notably, although most key TGFBR2 human TGFβ1-contact residues are conserved within the LBD[in] domains, the frog LBD[in] contains a non-conservative substitution at position 119 (E119L) within the C-terminal segment previously implicated in ligand interaction (Fig. 4c)[23]. Despite this substitution, the LBD[in] domain facilitates ligand binding, suggesting that the overall binding interface

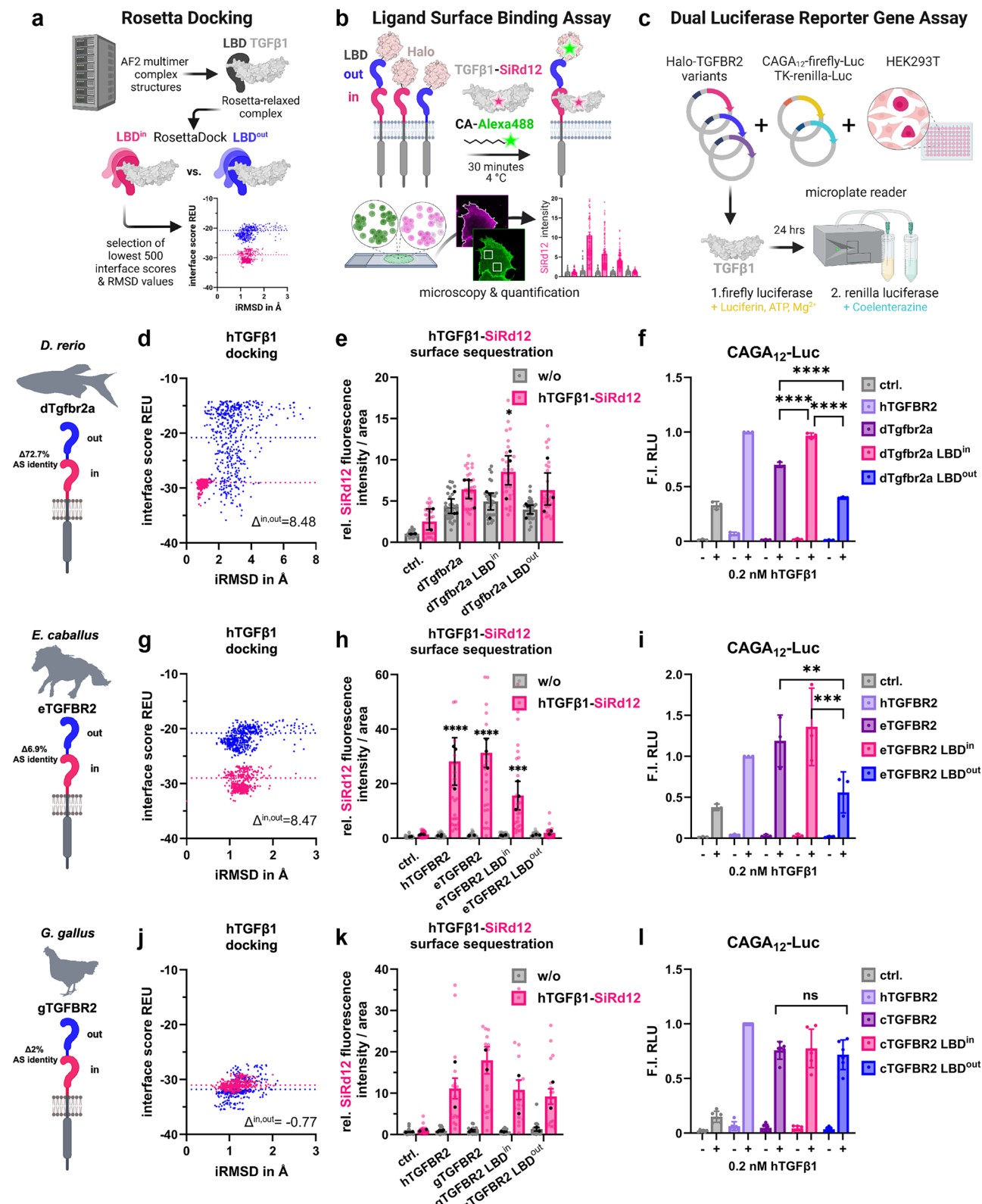

remains functionally preserved, potentially through compensatory interactions or structural tolerance within the receptor–ligand complex. Importantly, the presence of the LBD[out] does not impair binding or signaling function of the full length TGFBR2 for either species (Figs. 5h, i, and Supplementary Figs. 9e, f, 14, 15).

In contrast, chicken TGFBR2 shows a different pattern. Both LBD[in] and LBD[out] share a conserved TGFβ1 binding epitope and 98%

sequence identity, resulting in similar predicted and measured TGFβ1 binding. Each domain alone is sufficient to support efficient ligand binding and downstream signaling (Fig. 5j–l, Supplementary Fig. 16). Interestingly, the native chicken receptor containing both LBDs binds TGFβ1 more strongly than either single domain receptor although signaling output remains comparable regardless of the number of LBDs (Fig. 5k–l).

**Fig. 5 | TGFBR2 LBD duplications differently finetune signaling competence. a** Schematic representation of in silico binding analysis workflow using Rosetta docking. Alphafold (AF2 multimer) LBD-TGFβ1 complex structures are generated, followed by initial coordinate constrained relaxation to obtain optimized H-bonds. Docking of LBDs is then performed through RosettaScripts ($n = 2500$). Interface score (REU) versus iRMSD plots identify the top 500 models with the lowest scores, indicating different ligand binding capabilities. **b** Illustration of LSBA for visualization and quantification of fluorescent hTGFβ1 binding on COS-7 cells expressing respective Halo-tagged TGFBR2 receptor constructs. **c** Schematic representation of dual luciferase reporter gene assay used to assess TGFβ1 signaling activity through Halo-tagged TGFBR2 receptor orthologs. HEK293T cells are co-transfected with respective receptor variant, CAGA$_{12}$-firefly-Luc, and TK-renilla-Luc plasmids. Following TGFβ1 treatment for 24 h, Firefly luciferase activity (indicative of TGFβ1 pathway activation) and Renilla luciferase activity (normalization control) are measured. **d, g, j** in silico binding analysis via Rosetta docking of hTGFβ1 to inner (pink) and outer LBD (blue) of *D. rerio, E. caballus* and *G. gallus* depicted as interface score (REU) in relation to interface root mean square deviation (iRMSD) in angstroms. REU differences of LBD$^{in}$ and LBD$^{out}$ are calculated by respective mean REU values of each LBD. **e, h** hTGFβ1–SiR-d12 surface binding to *D. rerio* (**e**) and *E. caballus* (**h**) receptor variants, shown as relative fluorescence intensity per area ( ± hTGFβ1–SiR-d12). Data are mean ± SEM. $n = 3$ independent biological replicates (independent experiments; up to 10 cells per experiment averaged). Two-way

ANOVA with Dunnett's multiple comparisons test (two-sided; vs ctrl.) was used. Exact *P* values: (**e**) dTgfbr2a LBD$^{in}$ vs ctrl., $p = 0.0219$; (**h**) hTGFBR2 vs ctrl., $p < 0.0001$; eTGFBR2 vs ctrl., $p < 0.0001$; eTGFBR2 LBD$^{in}$ vs ctrl., $p = 0.0004$. (**k**) hTGFβ1–SiR-d12 surface binding to *G. gallus* receptor variants. Two representative independent experiments ($n = 2$) are shown; the experiment was repeated four times with similar results. Data are presented as mean ± SEM. No statistical testing was performed. **e, h, k** Error bars represent SEM; black dots indicate independent biological replicates, and grey and pink dots represent individual cells. **f, i, l** pSMAD2/3-sensitive CAGA$_{12}$ luciferase reporter activity of *D. rerio* (**f**), *E. caballus* (**i**) and *G. gallus* (**l**) receptor variants in the presence or absence of TGFβ1 (0.2 nM), shown as fold induction (F.I.) of relative light units (RLU) relative to hTGFBR2 (+TGFβ1). Data are mean ± SD. $n = 3$ independent biological replicates (independent experiments) for (**f, i**) and $n = 6$ independent biological replicates for (**l**). Two-way ANOVA followed by Tukey's multiple comparisons test (two-sided; within groups) for (**f**) or Dunnett's multiple comparisons test (two-sided; vs cTGFBR2 LBD$^{out}$ (**i**) or vs cTGFBR2 LBD$^{out}$ (**l**)), as appropriate, was used. Exact *P* values: (**f**) dTGFBR2a LBD$^{in}$ vs dTGFBR2a, $p < 0.0001$; dTGFBR2a LBD$^{out}$ vs dTGFBR2a, $p < 0.0001$; dTGFBR2a LBD$^{out}$ vs dTGFBR2a LBD$^{in}$, $p < 0.0001$; (**i**) eTGFBR2 vs eTGFBR2 LBD$^{out}$, $p = 0.0031$; eTGFBR2 LBD$^{in}$ vs eTGFBR2 LBD$^{out}$, $p = 0.0003$. Error bars represent SD. Source data are provided as a Source Data file. **a–d, g, j** Schematics created in BioRender. Trumpp, M. (https://BioRender.com/8p64dhh).

After characterizing the differential ligand-binding capacity of LBD-duplicated TGFBR2 variants, we sought to predict the impact of linker length between the two LBDs. While two identical LBDs may enable inter-ligand binding, thereby increasing total ligand engagement, a flexible linker of sufficient length could permit intra-ligand bivalency. In this scenario, two LBDs within a single receptor would occupy both type II receptor-binding sites, potentially interfering with the formation of a functional tetrameric receptor complex (Fig. 6a, and Supplementary Fig. 17a, b). To estimate the linker length required for intra-ligand bivalency, we modeled human TGFBR2 variants containing two identical LBDs connected by linkers of increasing length, starting from the 20-residue linker encoded at the E3–E2′ junction. Linkers of ≥36 residues were predicted to represent the minimal requirement for intra-ligand bivalency (Fig. 6b). Comparison with naturally occurring linker lengths across TGFBR2 variants revealed that most species possess linkers shorter than 24 residues, with the exception of *C. harengus* (44 residues) (Fig. 6c, and Supplementary Fig. 17c). While short linkers may permit inter-ligand binding, intra-ligand bivalency is therefore unlikely in most species. Notably, the membrane-distal LBD in *C. harengus* lacks key residues of the TGFβ-binding interface, consistent with reduced binding affinity (Supplementary Figs. 7, 11a), further arguing against intra-ligand bivalency.

Next, we engineered a synthetic human TGFBR2 variant with two identical LBDs which would result from E2-E3 duplication and assessed its ability to bind human TGFβ1. Consistent with the chicken receptor, LBD duplication significantly enhanced ligand binding, confirming the inter-ligand binding mode (Fig. 6d, e, and Supplementary Fig. 18). Whereas CAGA$_{12}$-luc activity after 24 h was comparable for both TGFBR2 variants (Fig. 6f), short term activation of SMAD2 was reduced in the presence of TGFBR2 2 LBD compared to TGFBR2 expression (Fig. 6g). Collectively, this highlights that 2 LBD TGFBR2 receptors remain functional but limit the downstream SMAD activation through excessive ligand binding (Fig. 6h).

These findings suggest that LBD duplication in TGFBR2 is a recurrent evolutionary mechanism capable of fine-tuning ligand binding without compromising signaling output, thereby offering a potential selective advantage under specific physiological or developmental contexts.

**LBD duplication generates low-signaling dTgfbr2a in zebrafish**
Notably, TGFBR2 two-LBD variants from different species exhibited different signaling capabilities, which raises the question how these

integrate into the critical fine-tuning of balanced BMP and TGFβ signaling in the respective species. To answer this, we analyzed publicly available RNA-seq datasets to screen for tissue-specific expression patterns in chicken, horse and zebrafish, which were characterized by different TGFBR2 splice variants (Supplementary Fig. 19a). In both chicken and horse, the TGFBR2 two-LBD variant was highly expressed in the cardiovascular and reproductive systems (Supplementary Fig. 19b–d). Interestingly, the high degree of amino acid homology between both LBDs is equally reflected in the coding sequence of exon 3 and exon 3′ for both TGFBR2 ortholog variants (Supplementary Fig. 19a). Exon duplication events have been associated with alternative splicing in multiple gene families, e.g., immunoglobulin super-family members and tropomyosin[24]. Indeed, in both chicken and horse, we observed single LBD variants due to alternative splicing but at a marginal rate, indicating that the TGFBR2 two-LBD variant is the predominant receptor (Supplementary Fig. 19b–d).

Similarly, zebrafish express only a two-LBD variant of dTgfbr2a and do not exhibit alternative splicing at this locus. However, unlike chicken and horse, zebrafish possess a second paralog, Tgfbr2b, which encodes a receptor variant containing a single LBD (Fig. 7a). To compare the functional properties of these paralogs, we assessed their ligand-binding affinity and signaling capacity. The LBD of dTgfbr2b displayed a modestly higher affinity for TGFβ1 relative to the inner LBD of dTgfbr2a (Fig. 7b). Notably, dTgfbr2b also showed markedly increased ligand sequestration at the cell surface, likely attributable to the inhibitory effect of the outer LBD in dTgfbr2a (Fig. 7c). This was reflected in reduced downstream signaling competence of dTgfbr2a *vs.* dTgfbr2b, suggesting potential for fine tuning TGFβ signaling by dTgfbr2a compared to dTgfbr2b, indicating that the two paralogs may enable nuanced regulation of TGFβ signaling in zebrafish (Fig. 7d).

To explore potential tissue-specific functions of the two zebrafish paralogs, we analyzed publicly available single-cell RNA-sequencing data from developing zebrafish embryos. In contrast to chicken and horse, where only the two-LBD variant is present, the single-LBD dTgfbr2b paralog was broadly expressed across all tissues (Fig. 7e). By contrast, the two-LBD dTgfbr2a paralog displayed selective enrichment in the intermediate mesoderm and hematopoietic lineages, with its expression increasing relative to dTgfbr2b during development (Fig. 7f, g).

Comparative transcriptomic profiling of cells expressing either Tgfbr2 paralog revealed largely non-overlapping gene expression signatures, suggesting distinct biological roles (Fig. 7h). Gene Ontology

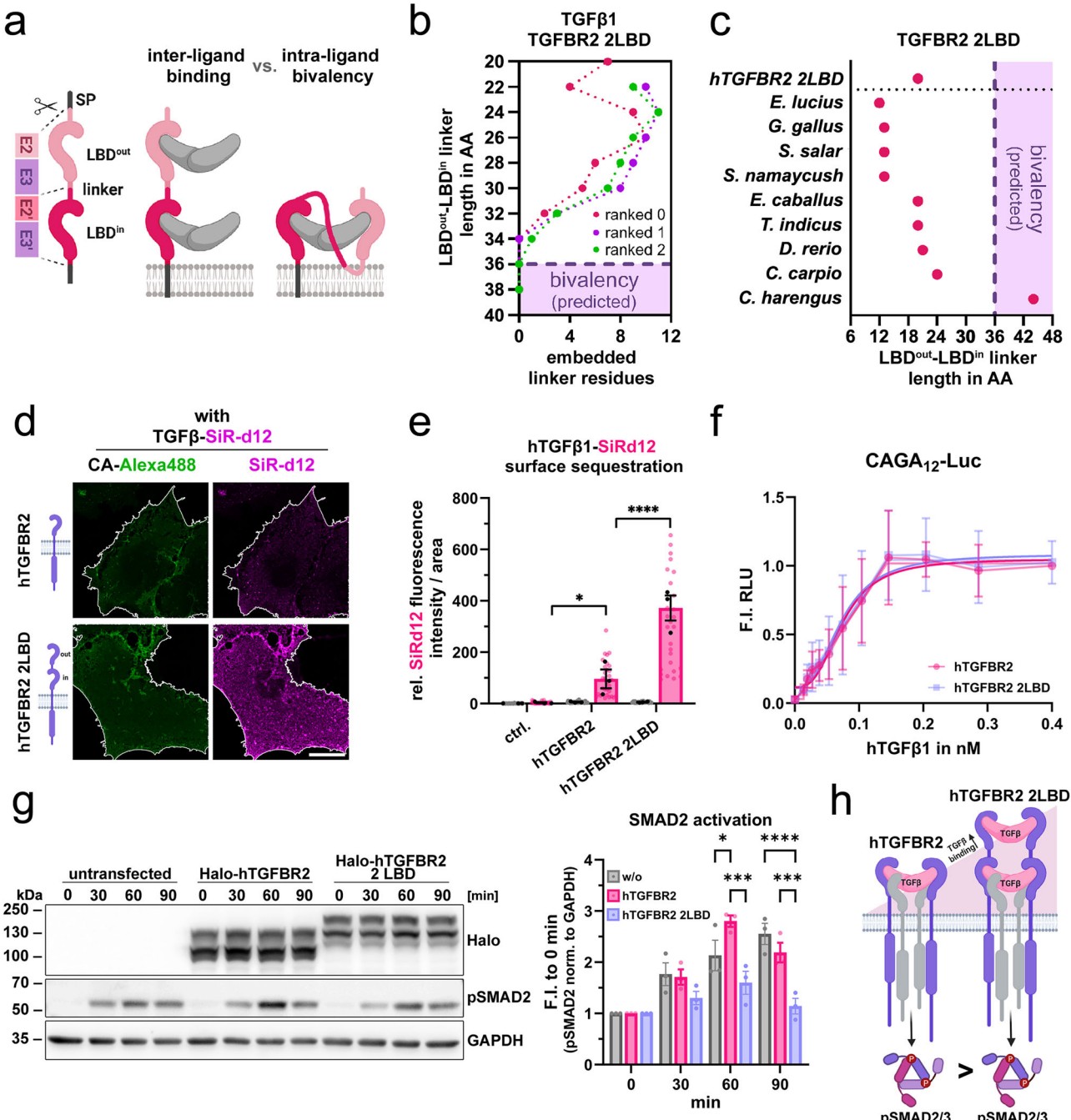

(GO) enrichment analysis revealed that genes specifically upregulated in dTgfbr2a$^+$ cells were predominantly associated with cytoskeletal remodeling, small GTPase signaling and migration, particularly within immune and cardiovascular contexts (Fig. 7i). Given the well-established role of TGFβ-signaling in driving cellular plasticity through epithelial-to-mesenchymal-transition (EMT)[25–28], we next interrogated the expression of canonical EMT transcription factors. Intriguingly, EMT drivers such as the orthologs of *Zeb1, Snail, Slug* and *Twist* were markedly enriched in dTgfbr2b$^+$ cells (Fig. 7j), consistent with its higher signaling competence relative to dTgfbr2a (Fig. 7d).

To validate these differences in vivo, we transiently expressed each receptor in developing zebrafish embryos. Expression of dTgfbr2b robustly induced phenotypes associated with imbalanced TGFβ/BMP-signaling[29,30], including increased incidence of cyclopia and ventralization compared to dTgfbr2a (Fig. 7k, and Supplementary Fig. 20). Notably, deletion of the outer LBD in dTgfbr2a restored

signaling activity, phenocopying the higher incidences of ventralization and cyclopia observed with dTgfbr2b, and indicating a structural basis for reduced signaling capacity of dTgfbr2a (Fig. 7k). Conversely, the isolated expression of dTgfbr2a LBD$^{out}$ enhanced cyclopia incidence relative to full-length dTgfbr2a, but not ventralization. Overall, the elevated ventralization observed in vivo is consistent with the higher signaling output measured in vitro for dTgfbr2a LBD$^{in}$ and dTgfbr2b (Fig. 5e, f; and Supplementary Fig. 12; Fig. 7c, d), suggesting that increased dTgfbr2 signaling capability contributes to the ventralizing effect (Fig. 7k).

Together, these findings reveal that LBD duplication in paralogous TGFBR2 receptors can drive neofunctionalization, yielding a receptor variant with attenuated signaling capacity. This structural innovation likely enables tissue-specific modulation of TGFβ-signaling output, offering a previously unrecognized mechanism by which vertebrates can diversify and fine-tune key developmental and homeostatic pathways.

**Fig. 6 | Human TGFBR2 LBD duplication enhances ligand binding without affecting downstream signaling. a** Schematic illustration of two potential modes of ligand binding by 2LBD receptors: inter-ligand binding and intra-ligand bivalency. Intra-ligand bivalency may require a sufficiently long and flexible linker between the two LBDs. Linker length is defined by the residues encoded at the junction of exons E3 and E2′. **b** Structural predictions assessing the minimal linker length required for intra-ligand bivalency using hTGFβ1 and a modeled hTGFBR2 2LBD. The number of linker residues embedded within the dimer interface is plotted against total linker length for the top three ranked models. Consistent across these predictions, a linker length of 36 residues did not result in threading through the dimer interior and was therefore defined as the predicted cutoff for intra-ligand bivalency. **c** Comparison of naturally occurring linker lengths between 2 LBDs across TGFBR2 variants from different species. Most variants fall below the predicted cutoff for intra-ligand bivalency, suggesting that intra-ligand bivalency is not feasible and that TGFβ1 binding could occur via an inter-ligand mode. **d** Representative confocal microscopy images of LSBA for hTGFβ1-SiR-d12 stimulated COS-7 cells expressing hTGFBR2 or hTGFBR2-2LBD. Scale bar ≙ 20 µm. **e** hTGFβ1−SiR-d12 surface binding to hTGFBR2 and hTGFBR2−2LBD, shown as relative fluorescence intensity per area ( ± hTGFβ1−SiR-d12). Data are mean ± SEM. $n = 3$ independent biological replicates (independent experiments; up to 10 cells per experiment averaged). Two-way ANOVA followed by Dunnett's multiple comparisons test (two-sided; vs hTGFBR2) was used. Exact P values: ctrl. vs hTGFBR2,

$p = 0.0404$; hTGFBR2−2LBD vs hTGFBR2, $p < 0.0001$. Error bars represent SEM; black dots indicate independent biological replicates, and grey and pink dots represent individual cells. **f** pSMAD2/3-sensitive CAGA$_{12}$ luciferase reporter activity of hTGFBR2 and hTGFBR2-2LBD in transfected HEK293 cells stimulated with increasing concentrations of hTGFβ1, shown as fold induction (F.I.) of relative light units (RLU) relative to control ( − hTGFβ1). Data are mean ± SD. $n = 3$ independent biological replicates (independent experiments). Error bars represent SD. Lines represent nonlinear regression fits. **g** (left) Immunoblot analysis of HEK293T expressing hTGFBR2 or hTGFBR2-2LBD, stimulated with 0.1 nM hTGFβ1 for 0–90 min and stained for Halo, pSMAD2 and GAPDH as loading reference, (right) Densitometric quantification of pSMAD2 levels normalized to GAPDH and expressed as F.I. relative to 0 min. Data are mean ± SEM. $n = 3$ independent biological replicates (independent experiments). Two-way ANOVA followed by Tukey's multiple comparisons test (two-sided; within groups) was used. Exact P values: (60 min) hTGFBR2 vs w/o, $p = 0.0257$; hTGFBR2−2LBD vs w/o, $p = 0.0857$; hTGFBR2−2LBD vs hTGFBR2, $p = 0.0001$; (90 min) hTGFBR2 vs w/o, $p = 0.2978$; hTGFBR2−2LBD vs w/o, $p < 0.0001$; hTGFBR2−2LBD vs hTGFBR2, $p = 0.0006$. Error bars represent SEM. **h** Summary scheme comparing hTGFBR2 and hTGFBR2-2LBD ligand binding and downstream pSMAD2/3 signaling. Increased ligand binding is not reflected on SMAD2/3 phosphorylation. Source data are provided as a Source Data file. **a, h** Schematics created in BioRender. Trumpp, M. (https://BioRender. com/8p64dhh).

## Discussion

In this study, we identify a striking evolutionary innovation: independent duplication of ligand-binding domains across three receptor classes (ACVR1, BMPR2, TGFBR2) in at least 12 vertebrate lineages. Our findings reveal that LBD multiplication serves as an evolutionary platform for modulating TGFβ-signaling at two distinct levels, by influencing ligand-binding capacity (e.g., chicken) or signal transmission efficacy (e.g., zebrafish). Moreover, we uncover recurrent instances of secondary LBD trimerization, notably within amphibians and salmonids, suggesting a sustained selective pressure for structural innovations in receptor design. These repeated, lineage specific adaptations point to a broader strategy by which vertebrates fine-tune pathway responsiveness through domain architecture. Given the association of TGFβ receptor mutations with numerous human developmental disorders[31–33], our findings underscore the evolutionary constraints on receptor core architecture, while simultaneously revealing unexpected structural plasticity through domain-level innovation.

The evolutionary roots of the TGFβ-receptors trace back to early metazoans such as sponges and placozoans, which encoded a minimal complement of two type I and one type II receptor genes[14]. With the emergence of bilateral symmetry, the TGFβ ligand repertoire expanded in parallel, reflecting the increasing demand for finely tuned BMP and TGFβ signaling during axis specification and early morphogenesis[34]. By contrast, vertebrates exhibit a markedly expanded TGFβ receptor repertoire, reflecting two rounds of whole-genome duplications, with additional lineage-specific duplications in teleosts, *Xenopus laevis*[35–37] and more recently in carps and salmonids[14,38,39]. This expansion supports the complex and context-dependent roles of TGFβ family ligands in vertebrate development, where they direct tissue specification, cell movement, and patterning along dorsal–ventral and anterior–posterior axes[7].

Despite this diversification, vertebrate genomes encode more ligands than receptors, making signaling specificity heavily reliant on selective ligand expression and combinatorial assembly of tetrameric receptor complexes[14,34]. Tight regulatory control is achieved through multiple mechanisms, including soluble antagonists, co-receptors, transcriptional modulators, and both post-translational and epigenetic modifications[40]. The evolutionary significance of maintaining these precise interactions is underscored by the strong sequence conservation observed in co-evolved ligand-receptor pairs[41].

Recurrent vertebrate WGDs generated additional TGFβ receptor paralogs, facilitating the evolutionary refinement of TGFβ signaling. Retention and divergence of paralogous receptor genes allowed one copy to evolve neofunctionalized or modular domain, while the other maintained a canonical receptor form[41,42]. This is exemplified by multiple teleost species, where Y-linked AMHR2 paralogs evolved truncated LBDs and ligand-independent activity, co-opting as sex-determining factors across diverse lineages, including sharks, catfishes, percids, the alligator pipefishes and the common seadragon[43–46].

Here we showed that twelve independent LBD expansions across *ACVR1, BMPR2*, and *TGFBR2* arose at different time points in vertebrate evolution. These events demonstrate that ligand-binding domain multimerization has repeatedly evolved as a novel layer of refining TGFβ receptor function.

The initial functional advantage of LBD duplication likely relates to the context-specific assembly of tetrameric TGFβ-receptor complexes[47]. While a second LBD does not alter receptor stoichiometry per se, it could enhance inter-ligand capture, especially for high affinity receptors such as TGFBR2, in tissues with low ligand availability or restricted diffusion. For instance, duplicated LBDs in chicken, tapir and salmon TGFBR2s are nearly identical, potentially facilitating enhanced ligand avidity or interaction with ECM-bound latent ligands, analogous to the role of the co-receptor endoglin in BMP9 signaling[48,49]. In these species the duplicated LBDs show low amino-acid substitution rates, in line with the clade-specific molecular clock, indicating that they are under strong evolutionary constraint and remain highly conserved. This is reflected by enhanced TGFβ1 binding of a synthetic human 2 LBD TGFBR2. Whereas, this receptor exhibits limited SMAD activation, likely due to high ligand sequestration, it remains an open question how 2 LBD receptors integrate into the native receptor complex compositions, including tetrameric complex stoichiometries and modulatory co-receptors. Notably, in lineages such as chicken and horse, alternative splicing can generate receptor variants that selectively include or exclude individual LBDs, suggesting that both duplicated domains remain functionally relevant and may be differentially deployed depending on cellular context.

This contrasts with zebrafish, where the duplicated outer LBD shows markedly higher amino-acid substitution rate, indicating relaxed constraint and suggesting that the domain is drifting or diversifying rather than being maintained for interchangeable function. Such relaxed constraint is likely enabled by the presence of the paralogous receptor (Tgfbr2b) that provides functional redundancy and permits neofunctionalization of the duplicated domains. Consistent with this, we show that the outer LBD of Tgfbr2a attenuates signaling, acting as an inhibitory module and illustrating domain-level neofunctionalization.

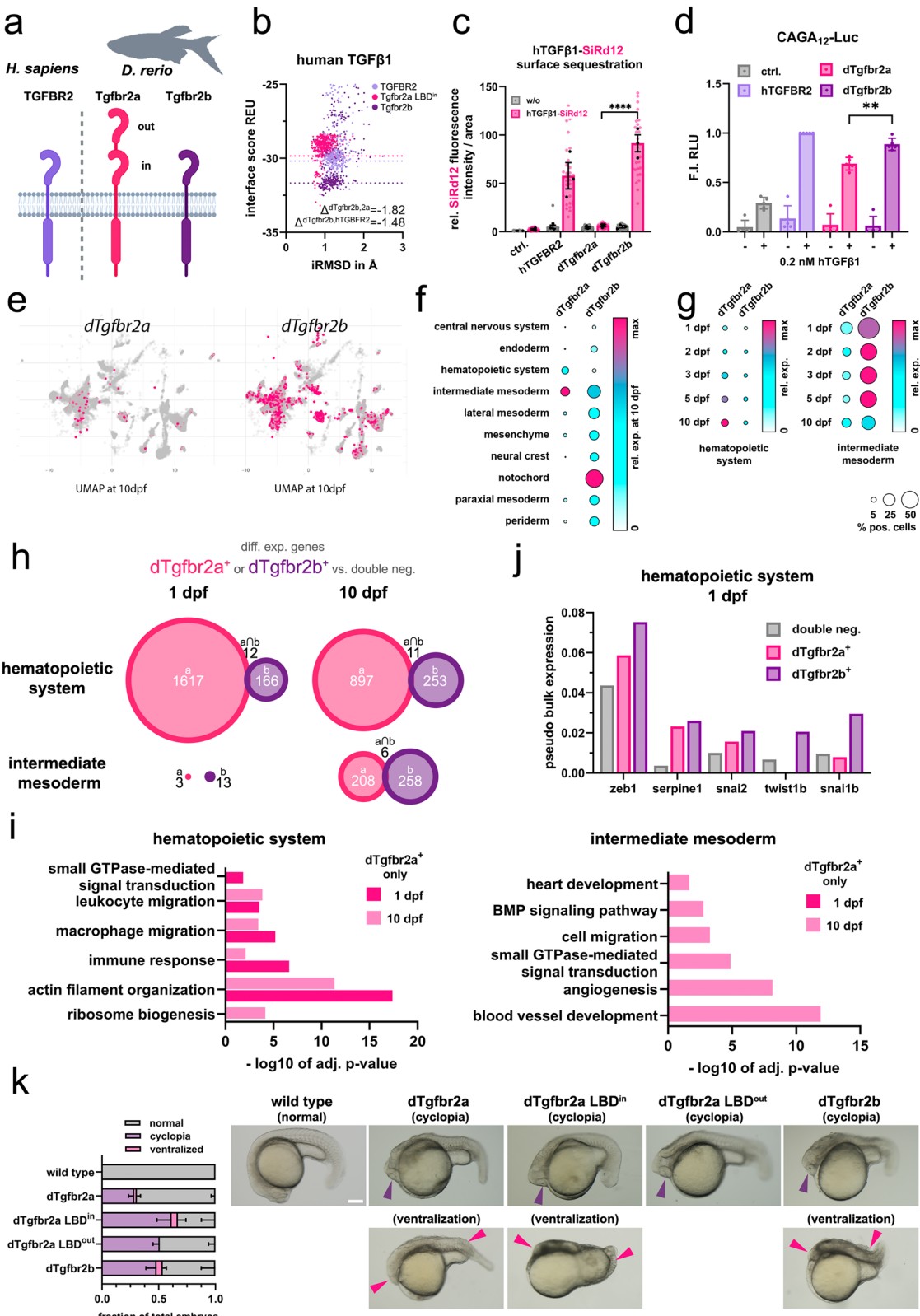

A remaining question is why LBD duplication has occurred repeatedly in only a subset of receptors while remaining rare across the broader TGFβ receptor family. Intriguingly, domain multimerization is not limited to vertebrates. Independent extracellular domain duplications in ancestral TGFBR2 orthologs have been described in invertebrates such as sponges and oysters[50,51], indicating that LBD duplication is an evolutionarily widespread strategy,

independent of WGDs. However, within vertebrates, sporadic occurrence of LBD duplication in only a subset of receptors, suggests strong structural constraints on evolvability, with the affected receptors uniquely tolerant of domain multiplication while preserving initial ligand binding/signaling integrity in certain organisms. Tolerance of LBD duplication in some receptors may be facilitated by the co-expression of functionally conserved paralogous partners, such as

**Fig. 7 | Duplicated LBD tgfbr2 paralog exhibits limited signaling competence and cell type specific expression pattern in zebrafish. a** Schematic comparison of human hTGFBR2, *Danio rerio* dTgfbr2a and dTgfbr2b receptor. **b** In silico binding analysis via Rosetta docking of hTGFβ1 to inner LBD of dTgfbr2a (pink), dTgfbr2b LBD (purple) and hTGFBR2 (lavender) depicted as interface score (REU) in relation to iRMSD in angstroms. REU differences are calculated by respective mean REU values of dTgfbr2b to dTgfbr2a or hTGFBR2. **c** hTGFβ1−SiR-d12 surface binding to hTGFBR2, dTgfbr2a and dTgfbr2b, shown as relative fluorescence intensity per area ( ± hTGFβ1−SiR-d12). Data are mean ± SEM. $n = 3$ independent biological replicates (independent experiments; up to 10 cells per experiment averaged). Two-way ANOVA followed by Dunnett's multiple comparisons test (two-sided; vs dTgfbr2a) was used. Exact $P$ values: hTGFBR2 vs dTgfbr2a, $p < 0.0001$; dTgfbr2b vs dTgfbr2a, $p < 0.0001$. Error bars represent SEM; black dots indicate independent biological replicates, and grey and pink dots represent individual cells. **d** pSMAD2/3-sensitive CAGA$_{12}$ luciferase reporter activity of hTGFBR2, dTgfbr2a and dTgfbr2b in the presence or absence of hTGFβ1 (0.2 nM), shown as fold induction (F.I.) of relative light units (RLU) relative to hTGFBR2 (+TGFβ1). Data are mean ± SD. $n = 3$ independent biological replicates (independent experiments). Two-way ANOVA followed by Dunnett's multiple comparisons test (two-sided; vs dTgfbr2b) was used. Exact $P$ values: dTgfbr2a vs dTgfbr2b, $p = 0.0018$. Error bars represent SD. **e** UMAP representation of *D. rerio* transcriptome at 10 dpf, highlighting dTgfbr2a and dTgfbr2b positive cells (pink). **f** relative expression of dTgfbr2a and dTgfbr2b at 10 dpf in respective tissue; size of circle represents percentage of positive cells (**g**) relative expression of dTgfbr2a and dTgfbr2b in hematopoietic system and intermediate mesoderm at different developmental stages; size of circle represents percentage of positive cells (**h**) Venn diagrams of differentially expressed genes in dTgfbr2a (pink) and dTgfbr2b (purple) positive cells compared to double-negative controls at 1 and 10 dpf in hematopoietic system and intermediate mesoderm (**j**) Pseudo bulk expression of known SMAD2/3 target genes in dTgfbr2a, dTgfbr2b positive and double negative cells of the hematopoietic system at 1 dpf (**i**) Go term analysis of dTgfbr2a positive cells at 1 and 10 dpf of the hematopoietic system and intermediate mesoderm. **k** Overexpression of dTgfbr2a, dTgfbr2a-LBD$^{in}$, dTgfbr2a-LBD$^{out}$ and dTgfbr2b in zebrafish embryos at 22 hpf. (left) Fraction of embryos displaying normal, cyclopic or ventralized phenotypes following overexpression of the indicated receptor constructs. For each independent experiment, fractions were calculated per phenotype category and subsequently averaged across experiments. Data are presented as mean ± SD. $n = 3$ independent biological replicates (independent experiments/clutches), with a minimum of 13 embryos analysed per condition in each experiment. Error bars represent SD. (right) Representative brightfield images of embryos exhibiting indicated phenotypes. Purple arrowheads indicate fused eye structure in embryos with cyclopia phenotype. Pink arrowheads point to the reduction of head and neural tube structures in ventralized embryos. Scale bar = 100 μm. Source data are provided as a Source Data file. **a** Schematics created in BioRender. Trumpp, M. (https://BioRender.com/8p64dhh).

ACVR1L alongside ACVR1 or ACVR2A/B alongside BMPR2. In contrast, TGFBR2 functions as the sole high-affinity type II receptor transmitting canonical TGFβ signaling. At the same time, the apparent restriction of LBD duplications to these receptors should be interpreted cautiously. Current genome assemblies and transcriptomic datasets remain uneven across the vertebrate phylogeny, and structural variants affecting ECDs may remain undetected in poorly sampled or fragmented genomes. Thus, additional examples of receptor domain multiplication may exist but fall below the current limits of detection.

Together, these findings reveal that LBD duplication acts as a versatile mechanism for modulating TGFβ-receptor function, enabling the evolution of context-sensitive signaling outputs. Given the pivotal role of TGFβ-signaling in regulating cell fate, plasticity, and migration, such domain-level adaptations likely contributed to the evolutionary diversification of vertebrate body plans. More broadly, these results establish LBD multimerization as a general mechanism to tune ligand engagement and signaling specificity within the TGFβ pathway.

## Methods

### Ethics and animal welfare statement
Our research complies with all relevant ethical regulations; the relevant animal protection committee of the Leibniz-IGB and the city of Berlin (LaGeSo) approved the animal use protocol where relevant. *Anguilla anguilla* and *Cyprinus carpio* were kept under animal husbandry permit ZH114 (LaGeSo, Berlin) at IGB; three other fish species (*Gnathonemus petersii, Pantodon buchholzi, Erpetoichthys calabaricus*) were obtained from commercial dealers and humanely euthanized using an overdose of buffered Tricaine PHARMAQ 1000 MG/G (MS222; concentration: 500 mg/L), for commercial fish at the day of arrival at IGB. This ensured that no pain, suffering, distress or lasting harm was inflicted on the animals. RNA samples of *Xenopus laevis* were obtained from control groups of an animal experiment approved by the German State of Health and Social Affairs (LaGeSo, Berlin, Germany; G0359/12).

Zebrafish experiments were performed in accordance with protocol BR22-1497 approved by the Institutional Animal Care and Use Committee (IACUC) of the National University of Singapore. Adult zebrafish were housed in recirculating aquaria systems at 28 °C under a 14 h/10 h light/dark cycle in the fish facility of the Department of Biological Sciences (DBS) at the National University of Singapore. Strain DBSWT zebrafish, source Department of Biological Sciences (DBS), National University of Singapore. Wild type zebrafish embryos were obtained by crossing corresponding adult male and female fish.

### RNA isolation and cDNA synthesis
Taxon sampling includes selection from *Danio rerio* (fin−adult, whole larvae), *Xenopus laevis* (brain, testis, liver and gonad mixture (adult), larvae), *Cyprinus carpio* (gonad, muscle, brain, heart, fin − adult), *Gnathonemus petersii* (gonad, muscle, brain, heart, fin − adult), *Pantodon buchholzi* (gonad, muscle, brain, heart, fin−adult), *Erpetoichthys calabaricus* (gonad, muscle, brain, heart, fin−adult), *Anguilla anguilla* (gonad, muscle, brain, heart, fin − adult), and *Gallus gallus* (DF1 cell line). Tissues of *Anguilla anguilla, Cyprinus carpio Gnathonemus petersii, Pantodon buchholzi, Erpetoichthys calabaricus* were stored in RNAlater (Thermo Fisher Scientific) for 24 h at 8 °C, then drained and stored at −80 °C until further processing. RNAs were extracted using TRIzol Reagent (Thermo Fisher Scientific, Waltham, USA) according to the supplier's recommendation and cleaned using the RNeasy Mini Kit (Qiagen). For synthesis of *Danio rerio* and *Gallus gallus* cDNA, 1 μg of total RNA was reversely transcribed by incubation with random primers (100 pmol μL$^{-1}$, Invitrogen) and M-MuLV reverse transcriptase enzyme (New England Biolabs) following manufactures recommendations. Synthesis of cDNA of *Xenopus laevis* and *Cyprinus carpio* was performed using iScript RT Supermix for RT-qPCR kit (Biorad); 10 μL of RNA (concentration of 80 ng/μL) were randomly reverse-transcribed following the manufactures recommendations.

### RNA sequencing and de novo transcriptome assembly
For transcriptomics of *Anguilla anguilla, Gnathonemus petersii, Pantodon buchholzi* and *Erpetoichthys calabaricus* muscle tissue, RNA and library processing and RNA-seq were carried out by NOVOGENE (Cambridge, UK) on a NovaSeq 6000 PE 150, generating 9.3 − 10.0 Gb of raw data per sample. Raw data of each sample was processed using the fq2fa −merge command from IDBA v1.1.1[52] package to convert fastq to interleaved fasta format. The corresponding files were assembled using idba_tran v 1.1.1 with default parameters. The highest kmer transcript assemblies (kmer 60) were aligned to a custom protein database (consisting of sequences for *ACVR1, ACVRL1, BMPR1A, BMPR1B, ACVR1B, ACVR1C, TGFBR1, ACVR2A, ACVR2B, TGFBR2, BMPR2, AMHR2* and *INHBA* from *Siniperca chuatsi*[53] by blat[54] using parameters (−t=dnax −q=prot). Blat results were sorted by matching score and the best hit transcripts were written to a bed6 formatted table and extracted from the transcriptomes in 5' to 3' orientation using bedtools getfasta (-name -s)[55]. The extracted sequences were further processed as described below.

## Phylogenetic analysis

Sequences of TGFβ receptor family orthologs were screened for LBD multimerization using Ensembl gene tree function[56] and NCBI blast (https://blast.ncbi.nlm.nih.gov/Blast.cgi) against available vertebrate transcriptomes and/or genomes. RNA-seq was performed for selected species (*Gnathonemus petersii, Pantodon buchholzi, Erpetoichthys calabaricus, Anguilla anguilla*) to complement missing information in respective taxa. In the absence of available transcriptome data, ORFs were predicted in regions identified by nucleotide Blast of closely related species using Augustus web interface (https://bioinf.uni-greifswald.de/augustus/)[57]. LBDs were predicted from amino acid sequences using NCBI Conserved Domain Search web interface (https://www.ncbi.nlm.nih.gov/Structure/cdd/wrpsb.cgi)[58]. Phylogenetic trees were extracted from NCBI taxonomy CommonTree (https://www.ncbi.nlm.nih.gov/Taxonomy/ *CommonTree/wwwcmt.cgi*) and represented using Interactive Tree of Life (iTOL) v6[59]. Species images were extracted from Adobe Stock (ID: #301044215, #327193716, #370603116, #61858943, #115805136, #490808539, #125573996, #730691035, #552043292, #227759813, #574310862, #35897212, #370612679) under the license of the "Max Planck Institut für Molekulare Genetik, Berlin" and edited in Adobe Photoshop Version 26.8.0.

## RNA splicing analysis

To analyse *TGFBR2* alternative splicing in horse and chicken, RNA-seq data from various tissues (horse: PRJNA1017964; chicken: PRJEB26695) were aligned to the respective reference genomes (chicken: GCA_016699485.1; horse: EquCab3.0) using STAR v2.7.9a. Unique and multi-mapping reads for *TGFBR2* exons 1–5 were extracted from the SJ.out.tab STAR output files using standard Python scripts. For sashimi plots, merged BAM files were visualized in IGV, and coverage tracks were exported. Junction reads were manually corrected by replacing them with unique and multi-mapping counts from the SJ.out.tab files. Reported values correspond to the sum across all analysed replicates. In chicken samples, no reads corresponding to the E1–E2′ junction were detected. In horse, E2 inclusion relative to E2′ was calculated using uniquely mapping reads only. For the final table, only samples with at least 40 unique reads for the constitutive E4–E5 junction and tissues with at least two biological replicates were included. For horse, selected tissues known to be regulated by TGFβ-signaling are represented.

## RNA single cell analysis in zebrafish

Single-cell RNA-seq data from the zebrahub atlas were downloaded and processed in R using Seurat (v4)[60]. Cells with <500 detected features or genes expressed in <30 cells were excluded. Data were normalized, variable features identified, and expression values scaled prior to PCA (50 components) and UMAP embedding (first 30 PCs). Metadata annotations from the zebrafish anatomy ontology available from the zebrahub atlas were used for cell-type assignments. Gene expression was visualized on UMAPs and dot plots. For the intermediate mesoderm and hematopoietic system, cells were stratified into four groups based on tgfbr2a/tgfbr2b expression (double-positive, tgfbr2a + , tgfbr2b + , or double-negative). Differential expression relative to double-negative cells was assessed using Seurat's FindMarkers function, and pseudobulk-like expression profiles were generated by using Seurat's AverageExpression function. Differentially expressed genes were filtered by a log2 FC $\geq 0.58$ and an adjusted $p$ value $\leq 0.05$. Significantly diff. exp. genes relative to double-negative condition were compared using DeepVenn[61] (https://www.deepvenn.com/) and visualized in Photoshop. gProfiler g:GOSt (https://biit.cs.ut.ee/gprofiler/gost) was used for functional enrichment analysis of genes specifically enriched in dTgfbr2a+ or dTgfbr2b+ cells compared to double-negative cells on indicated time points.

## Molecular cloning

To functionally characterize TGFBR2 or BMPR2 receptors of *Danio rerio, Cyprinus carpio, Xenopus laevis, Gnathonemus petersii, Gallus gallus* and *Equus caballus*, we generated an N-terminally Halo-tagged receptor expression plasmid library (Supplementary Table 1). For this, we used various cloning methods including restriction-, Gibson-, blunt end- and deletion mutagenesis cloning, specified for each animal below. All primers used for PCR are listed in (Supplementary Table 2). Cloning PCRs were carried out on a Peltier Thermal Cycler PTC-200. The elongation time was adapted according to the product size (Phusion Pol. = 1 kb/min). PCR products were resolved by agarose gel electrophoresis and purified using NucleoSpin Gel and a PCR Clean-up kit (Macherey-Nagel), according to the manufacturer's guidelines. Gibson cloning using NEBuilder HiFi DNA Kit (New England BioLabs) was performed according to the manufacturer's protocol. Final products were transformed in chemically competent DH5α *E. coli* bacteria cells. All constructs were confirmed via Sanger sequencing followed by whole plasmid sequencing. Expression is confirmed via Western blot (Supplementary Fig. 21).

**Danio rerio and cyprinus carpio.** drTgfbr2a, drTgfbr2b and cTgfbr2a were amplified from respective cDNA and subcloned into a previously generated pcDNA3.1. hTGFBR2-halo plasmid[21] in between *EcoRI* and *NotI* sites, replacing the original receptor ORF resulting in drTgfbr2a-Halo, drTgfbr2b-Halo and cTgfbr2a-Halo expression plasmids. Further, to generate drTgfbr2a-LBD^in-Halo, drTgfbr2a-LBD^out-Halo, cTgfbr2a-LBD^in-Halo and cTgfbr2a-LBD^out-Halo constructs, deletion mutagenesis was performed on drTgfbr2a-Halo and cTgfbr2a-Halo templates. Deletion primers were used to flank either the inner or outer ligand binding domain sequences. Subsequently, PCR constructs were ligated via blunt end ligation.

**Gnathonemus petersii.** Following the RNA-seq analysis of *Gnathonemus petersii* heart tissue, we ordered the synthesis of gnBmpr2a-pTwist-CMV plasmid at "Twist Bioscience" and amplified gnBmpr2a, gnBmpr2a-LBD^mid-in and gnBmpr2a-LBD^in. We then subcloned these constructs into hTGFBR2-Halo plasmid in between *EcoRI* and *NotI* sites, replacing the original receptor ORF resulting in gnBmpr2a-Halo, gnBmpr2a-LBD^mid-in-Halo and gnBmpr2a-LBD^in-Halo expression plasmids. Further, to generate gnBmpr2a-LBD^out-Halo and gnBmpr2a-LBD^mid-Halo constructs, Gibson cloning was performed on gnBmpr2a-Halo template.

**Xenopus laevis.** XlTgfbr2L was amplified from respective cDNA and subcloned using Zero Blunt™ PCR Cloning Kit (Thermo Fisher Scientific) according to the manufacturer's protocol into a pCR™-Blunt-vector. The full length xlTgfbr2.L-Halo construct was synthesized by "Genewiz". Consecutively, Gibson-cloning was performed on this vector. By this, we obtained xlTgfbr2.L-LBD^in-Halo, xlTgfbr2.L-LBD^out-Halo, xlTgfbr2.L-LBD^mid-in-Halo.

**Equus caballus.** Due to high sequence identity between the inner and outer ligand binding domains, we obtained a codon optimized eTGFBR2-pTwist-CMV plasmid from "Twist Bioscience" and amplified eTGFBR2. We then subcloned this construct into hTGFBR2-Halo plasmid in between *EcoRI* and *NotI* sites, replacing the original receptor ORF resulting in eTGFBR2-Halo. Further, to generate eTGFBR2-LBD^in-Halo and eTGFBR2-LBD^out-Halo constructs, Gibson cloning was performed on eTGFBR2-Halo template.

**Gallus gallus.** gTGFBR2 (ENSGALT00000018657.6) and gTGFBR2-LBD^out (ENSGALT00000037691.5) were amplified from respective cDNA and subcloned into hTGFBR2-halo plasmid in between *EcoRI* and *NotI* sites, replacing the original receptor ORF resulting in gTGFBR2-

Halo and gTGFBR2-LBD$^{out}$-Halo. The subcloned gTGFBR2 ORF contained an additional exon in between the kinase coding exon 6 and exon 7, which was removed by Gibson cloning. Thereby, both gTGFBR2 and gTGFBR2-LBD$^{out}$ variants encoded for the same kinase, making a direct comparison of the duplicated LBD signaling competence possible. Further, to generate gTGFBR2-LBD$^{in}$-Halo, deletion mutagenesis was performed on gTGFBR2-Halo template. Deletion primers were used to flank the outer ligand binding domain. Subsequently, PCR constructs were ligated via blunt end ligation.

## Capped mRNA synthesis and microinjection in zebrafish

To synthesize capped mRNA, the mMESSAGE mMACHINE T7 Transcription Kit (ThermoFisher) was used with *DraIII*-linearized dTgfbr2a-Halo, dTgfbr2a LBD$^{in}$-Halo, dTgfbr2a LBD$^{out}$-Halo, and *BfuI*-linearized dTgfbr2b-Halo plasmids as templates, respectively. To synthesize capped *PMT-mEGFP* (membrane bound EGFP) mRNA, the mMESSAGE mMACHINE SP6 Transcription Kit (ThermoFisher) was used with *NotI*-linearized *pcs2+-PMT-mEGFP* [62,63] plasmids as templates. The final concentrations of synthesized mRNAs were measured by Qubit 4 Fluorometer (ThermoFisher) using Qubit™ RNA HS Assay Kit (ThermoFisher). For microinjection, wild type (WT) embryos were collected immediately after spawning and transferred to 0.3X Danieau's solution (17.4 mM NaCl, 0.21 mM KCl, 0.12 mM MgSO$_4$, 0.18 mM Ca(NO$_3$)$_2$, 1.5 mM HEPES, pH = 7.2). Injected embryos were raised in a 28 °C incubator until 22 hpf for screening and imaging. Embryonic stages were defined by hours post fertilization (hpf) at 28 °C and morphological features [64]. To compare the respective effects of dTgfbr2a-Halo, dTgfbr2a LBD$^{in}$-Halo, dTgfbr2a LBD$^{out}$-Halo and Tgfbr2b-Halo in WT embryos, 170 pg *dTgfbr2a-Halo*, 148 pg *dTgfbr2a LBD$^{in}$-Halo*, 148 pg *dTgfbr2a LBD$^{out}$-Halo*, or 148 pg of *dTgfbr2b-Halo* capped mRNAs were co-injected with 10 pg *PMT-mEGFP* at the border between cytoplasm and yolk of 1-cell stage WT embryos, respectively.

## Fluorescence screening and brightfield imaging of dTgfbr2-Halo injected zebrafish embryos

Injected embryos were incubated in a 28 °C incubator and screened for mEGFP fluorescence at 22 hpf. mEGFP-positive embryos were characterized into three categories (normal, cyclopia and ventralized) based on eye, head and trunk morphologies for quantification. Representative embryos were imaged under a stereomicroscope (Nikon, SMZ18).

## Cell culture

COS-7 and HEK293T cells were obtained from the German Collection of Microorganisms and Cell Cultures (DSMZ) and cultured in Dulbecco's Modified Eagle's Medium (DMEM) supplemented with 10% FCS, 2 mM L-glutamine and penicillin (100 units/mL) / streptomycin (100 µg/mL) (DMEM full medium) in a humidified atmosphere at 37 °C and 5% CO$_2$ (v/v). COS-7 and HEK293T cells were maintained in T175 flasks and split 1:5 or 1:10, depending on need and were kept sub-confluent. For passaging, cells were washed once with PBS before being removed from the flasks surface with trypsin/EDTA (0.05/0.02% in PBS).

## SDS-PAGE & Western-blotting

For sodium dodecyl sulphate polyacrylamide gel-electrophoresis (SDS-PAGE), cells transiently expressing Halo-receptors were lysed in 150 µL Laemmli buffer and frozen at −20 °C. To ensure a homogeneous loading, cell lysate was pulled through a 1 ml syringe and boiled for 10 min at 95 °C before loading onto 10% polyacrylamide gels. After performed gel-electrophoresis, proteins were transferred onto Methanol-activated PVDF membranes by Western-blotting. Next, membranes were blocked for 1 h in a solution containing 0.1% TBS-T and 3% w/v bovine serum albumin (BSA), then washed three times in 0.1% TBS-T and incubated with indicated primary antibodies overnight at 4 °C. Primary antibodies: anti-Halo (ProMega; #G9211; monoclonal

mouse antibody), anti-pSMAD2 Ser465/467 (Cell Signaling; #3108; monoclonal rabbit antibody), and anti-GAPDH (Cell Signaling; #2118; monoclonal rabbit antibody) were used at a 1:1000 dilution in 3% w/v BSA/ TBS-T solution. For HRP-based detection, membranes were incubated with secondary goat-α-mouse or goat-α-rabbit IgG HRP conjugates (± 0.8 mg/ml, Dianova; #111-035-144, #115-035-068) for 1 h at a dilution of 1:10000. Chemiluminescent reactions were processed using WesternBright Quantum HRP substrate (Advansta Inc.) and documented on a FUSION FX7 digital imaging system.

## Confocal microscopy & ligand surface binding assay (LSBA)

LSBA assay and consecutive image analysis and quantification was previously described in detail for SiR-d12-hTGFβ1 and Cy5-hActivin A [21]. In brief, for visualization of SiR-d12-hTGFβ1 and Cy5-hActivin A ligand binding, 200.000 COS-7 cells / well were seeded in 1 mL DMEM full medium on glass cover slips in a 12 well plate. On the following day, cells were transfected with desired constructs (500 ng DNA per well) using Lipofectamine2000 according to the manufacturer's instructions. The day after, cells were washed with PBS and, additionally to 0.5 mM fluorescent HaloTag-ligand CA-Alexa488 (Alexa Fluor 488, Promega, #G1002), simultaneously incubated with saturating conc. of SiR-d12-hTGFβ1 or Cy5-hActivin A for 30 min at 4 °C. Subsequently, cells were washed once with ice-cold PBS before fixation with 100% methanol for 5 minutes. Cells were washed again with PBS and mounted with Fluoromount G (Invitrogen, 00-4958-02). Confocal images were acquired on a commercial expert line Abberior STED microscope, using the Imspector software from Abberior Instruments (Version 16.3) in line-scanning confocal mode; objective lens: 100X NA1.4 (oil) [UPLS], pinhole: 1.0 AU, range: 75 µm × 75 µm, confocal pixel size: 60 nm, pixel dwell time: 5/10 µs. Images were acquired using 485 nm (20% laser power) and 640 nm excitation (20% laser power). The detection windows were set to 500−540 nm and 650−750 nm. Confocal raw data were post-processed and adjusted for color and contrast (linear adjustments maintained for confocal datasets represented within one figure) using Fiji (ImageJ) software and Adobe Photoshop (Adobe Systems). Surface binding quantification of SiR-d12-hTGFβ1 and Cy5-hActivin A on COS-7 cells transiently expressing Halo-Receptor constructs was performed with Fiji. Per cell, four regions of interest (ROI) (100 µm²) were chosen, and the raw integrated intensity (RawIntDen) of each ROI was measured both in receptor and ligand channels. Per condition, 30 cells were quantified in total in 3 independent experiments ($N = 3$). Where fewer than three datasets are shown for clarity, the images are representative; all replicates exhibited consistent trends despite variability in signal intensity. Receptor-ligand binding was calculated relative to intensity values of untransfected COS-7 cells, representing endogenous ligand-receptor binding. Normalized RawIntDen values of receptors and ligands were plotted in GraphPad Prism 10.5 (GraphPad Software Inc.).

## Dual luciferase reporter gene assay

For dual luciferase reporter gene assay, 50.000 HEK293T cells / well were seeded in 200 µL DMEM full medium in a 96 well plate. On the following day, cells were transfected with 50 ng of corresponding Halo-tagged receptors constructs together with 50 ng of SMAD2/3 sensitive (CAGA)$_{12}$ luciferase reporter. A constitutively expressing construct encoding renilla luciferase (RL-TK; Promega) was co-transfected (30 ng) as internal control. The next day, cells were starved in serum-free medium for 3 h before stimulation with 0.2 nM rhTGFβ1 (PeproTech, Hamburg, Germany) or 0.02 nM rhActivin A (Gift from Marko Hyvönen) for 24 h. Cell lysis was performed using passive lysis buffer (Promega, # E1910) and measurement of luciferase activity was carried out according to manufacturer's instructions using a TECAN Spark plate reader. Experiments testing the TGFBR2 ortholog signaling competence were normalized against the rhTGFβ1 response

of hTGFBR2-Halo. All experiments were performed at least three independent times.

## Structure prediction and interface energy calculations

The structure of the receptor and ligand complexes were first predicted using AlphaFold2 multimer[65,66] using the single receptor ECDs and two ligand chains (either hActivin A, hTGFβ1 or the respective animal variants) for the ligand homodimers. In order to obtain a physically more validated structure while simultaneously scoring the interface, first coordinate constrained relaxation protocol was performed to optimize H-bonds based on the Rosetta Energy Function 2015 (ref2015_cst) within RosettaScripts. Local Monte-Carlo-based protein-protein docking using RosettaScripts[20] was performed in 2500 replicas. Here, first, the "Docking" mover was used with low resolution (backbone plus centroid) flags before a high resolution (full atom) docking (using ref2015 scoring function). Other parameters as distance perturbation and angle perturbation were left on default. The Interface RMSD (iRMSD) was plotted against the score of the interface residues (i_SC) of the top 500 structures (determined by i_SC) and mean binding energies were calculated. Lower interface score correlates with favourable interface energies. Lower iRMSD correlates to docking structures closer to the starting AlphaFold2-multimer conformation. For structural representation, the top 10 hits showing the lowest i_SC were structurally investigated for the final homology model.

To evaluate whether linker length permits intra-ligand bivalency between two LBDs of hTGFBR2 2LBD, complexes with hTGFβ1 dimer and varying linker lengths were generated using AF2 as described above. In the predicted structures, both receptor domains are placed at their canonical binding positions on opposing sides of the TGFβ dimer, with the interdomain linker spanning between them. To assess whether the linker is long enough to wrap around the dimer surface, or whether it is too short and forced to thread through the dimer interior, we developed a spatial embedding metric. For each linker residue, identified by matching the known core sequence (SESVNNDMIVTDNNGAVKFP) and its flanking GS extensions, the surrounding space was partitioned into eight octants. A residue was classified as embedded within the dimer if heavy atoms of the ligand occupied five or more of eight octants within 5 Å of its Cα atom. AF2 generated structures with misplaced receptor domains were excluded. A linker length of 36 residues did not result in threading through the dimer interior and was therefore defined as the predicted cutoff for intra-ligand bivalency. This threshold was derived from the top three ranked predictions. Naturally occurring linker lengths in TGFBR2 variants were compared to this cutoff. All used scripts for structure predication and interface energy calculations are available under (https://github.com/agknaus/Jatzlau-Trumpp-et-al.-structural-computational-analysis/).

## Molecular clock and evolutionary rate analysis of LBDs

Amino-acid sequences of LBD regions were aligned in MEGA X[67]. Alignments were generated using the MUSCLE algorithm with default parameters. Pairwise evolutionary distances between LBD sequences were calculated in MEGA X using the Analyze → Compute Pairwise Distances function. Distances were estimated under the Jones–Taylor–Thornton (JTT) substitution model with gamma-distributed rate heterogeneity (Γ = 1.0) and a homogeneous substitution pattern among lineages. The substitution type was set to Amino acid, and gaps or missing data were handled using pairwise deletion. Variance estimation was performed using the bootstrap method with 1000 replicates. For each focal species (chicken, horse, salmon, zebrafish), distances were calculated between its inner LBD and the orthologous inner LBDs of other species within the same clade. Distances between the two LBD copies within the focal species were computed separately to quantify intra-species divergence associated with the domain duplication event.

Species divergence times for all pairwise comparisons were obtained directly from TimeTree.org[68]. These values were used as the independent variable in all molecular-clock analyses. For each clade, the amino-acid substitution rate (r, substitutions per site per million years) was estimated by performing a weighted linear regression of pairwise amino-acid distances against their corresponding TimeTree-derived species divergence times. The slope of the regression provided the clade-specific molecular-clock rate for the inner LBD domain. The observed amino-acid distance between the two LBD copies within each focal species was divided by the clade-specific molecular-clock rate to obtain the duplication-specific evolutionary rate ($r_{(dup)}$). This value reflects the effective rate of amino-acid substitution between the two paralogous domains since their duplication.

## Statistical analysis

All statistical tests were performed using GraphPad Prism version 10.5.0 software and are listed in the figure legends. Normal distribution of data sets was tested with the Shapiro–Wilk normality test. In cases of failure to reject the null hypothesis, the ANOVA and Tukey's or Dunnett post hoc test were used to check for statistical significance under the normality assumption. For all experiments statistical significance was assigned, with an alpha-level of $p < 0.05$.

## Graphical schemes and figures

Graphs were generated using GraphPad Prism 11.0.1. Molecular structure representations were generated using PyMOL Version 3.1.0. Selected schematic elements were created with BioRender.com under a publication license and incorporated into the final figure panels during assembly in Adobe® Photoshop (Adobe Systems, San José, USA). BioRender-created schematic elements are included in the graphical legend elements of Figs. 1 and 2; Fig. 3a, e; Fig. 4a; Fig. 5a–c, d, g, and j (left panel); Fig. 6a, h; Fig. 7a; Supplementary Fig. 1a, b; Supplementary Fig. 2a, b; Supplementary Fig. 4b–e; Supplementary Fig. 9a,d (left); Supplementary Fig. 10a–e (receptor scheme); Supplementary Fig. 11a–d (left); Supplementary Fig. 17a,b; and Supplementary Fig. 19a,c,d. These elements are covered by the BioRender publication license: Created in BioRender. Trumpp, M. (https://BioRender.com/8p64dhh).

## Reporting summary

Further information on research design is available in the Nature Portfolio Reporting Summary linked to this article.

## Data availability

The RNA raw sequence datasets for *Gnathonemus petersii, Pantodon buchholzi, Erpetoichthys calabaricus* and *Anguilla anguilla* generated in this study have been deposited in the NCBI/SRA under the BioProject database under accession code PRJNA1314022. Source data are provided with this paper.

## Code availability

All used scripts for structure predication and interface energy calculations are available under https://github.com/agknaus/Jatzlau-Trumpp-et-al.-structural-computational-analysis/, https://doi.org/10.5281/zenodo.19291097[69].

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

## Acknowledgements

The authors would like to thank Francesca Bottanelli (Freie Universität Berlin) for providing access to her commercial expert line Abberior STED microscope.

## Author contributions

J.J., M.T. designed the study, evaluated the data and generated all figures; M.T., J.J., J.K., Y.L., W.B., H.B. performed the experiments; J.J and M.T. performed the phylogenetic analysis; L.O., J.K. performed receptor homology modeling & docking calculations; M.T. produced hTGFβ1-SiR-d12; Y.L. performed zebrafish experiments; M.S. collected the tissue samples of *G. petersii, P. buchholzi, E. calabaricus, A. anguilla* and *X. laevis*, and prepared RNA and cDNA and discussed ancient/recent genome duplications; C.W. provided *D. rerio* cDNA samples. M.T., J.J., J.K. and H.B. cloned all Halo-receptor constructs. H.K. processed the transcriptome data and analyzed the results. M.P. performed RNA splicing analysis, P.M. analyzed scRNA-seq data; J.J. W.B. analyzed LSBA data; P.K., M.S., C.W. and S.M. discussed the project and gave critical input. J.J., M.T. and P.K. wrote the manuscript; all authors commented on the manuscript.

## Funding

J.J. and P.K. acknowledge the support from Deutsche Forschungsgemeinschaft DFG (SFB1444) and the Einstein Center ECRT). M.T. and L.O. were supported by the Max Planck Research School IMPRS-BAC. Funding has been in part provided by the German Research Foundation (DFG) STO493/8-1 to MS. C.W. was funded by grants from the Singapore Ministry of Education (grant numbers MOE-2016-T3-1-005 and MOE-T2EP30221-0008). W.B. was supported by Berlin–Brandenburg School for Regenerative Therapies. Open Access funding enabled and organized by Projekt DEAL.

## Competing interests

The authors declare no competing interest.
