## [Peer Review file · Nature Communications]

Recurrent evolution of ligand-binding domain multiplicity fine-tunes TGF β signaling in vertebrates

Corresponding Author: Professor Petra Knaus

Version 0:

Reviewer comments:

Reviewer #1

(Remarks to the Author)

The manuscript by Jatzlau and co-workers is focused on building upon the prior observation by the same group that in medaka (Japanese rice fish), a single exon coding for the TGF- β family type I receptor ACVR1 underwent triplication to produce a receptor with three tandem ligand binding ECDs.

Through analysis of genomic sequence information of related fish, and other species, such as frogs, horses, and chickens, the authors have identified 12 more instances of ECD multiplication in three TGF- β family receptors, the type I receptor ACVR1 (and a close paralog ACVR1L) and the type II receptors BMPR2 and TGFBR2 (and its paralog TGFBR2b), which for each ECD are encoded by two exons.

Through sequence analysis, modeling of the complexes of BMPR2 and TGFBR2 with their high affinity ligands, ActA and TGF- β 1, respectively, as well as cell surface fluorescence-mediated Ligand Binding Surface Assays (LBSA assay), and reporter assays, the authors generally find that the inner-most ECD is the one that is most highly conserved, in terms of its ligand binding residues, relative to the known human-ligand complex structure, and the one that has the greatest binding affinity and signaling activity – conversely, the more distal domains have lower conservation of interacting residues, lower surface affinities, and lower signaling activity. Interestingly though, there are some exceptions to this, for example in chickens, the outmost TGFBR2 ECD is essentially just as conserved as the innermost ECD and its found that although the tandem ECD confers high surface affinity, the signaling output is nonetheless comparable to that of a single ECD. In zebrafish, it was found that there are two TGFBR2 genes, which are differentially expressed, TGFBR2b, which has a single, but evidently higher affinity ECD, and TGFBR2a, which has tandem ECDs, with the outer ECD attenuating the ligand binding and signaling output. In developing zebrafish embryos, the more actively signaling TGFBR2b variant, or the TGFBR2a variant with the outer ECD deleted, were characterized by increased ventralizing activity, suggesting that the two TGFBR2 genes may have evolved to fine tune TGF β signaling to enable different roles in vivo.

The authors conclude that ECD duplication can enable neofunctionalization (and thus tuning of signaling), which is indeed an interesting observation that has not previously been appreciated for TGF- β family signaling.

Overall, while I think this is an important finding that merits publication, there are nonetheless a few major and several minor questions that I think should be addressed prior to publication – one of the major questions relates to the broader significance of these findings – that is if ECD duplication is enabling neofunctionalization, and presumably with this advantages to the organism, why is ECD duplication not been more fruitful in evolution (i.e. an adaptation that has become more widespread as organisms have diversified)? Another major question is why has this occurred in just these genes – why not also in other related type I and type II receptors of the TGF- β family?

Other more minor (but nonetheless important to address) questions are as follows:

1. In the studies comparing zebrafish TGFBR2b and TGFBR2a in zebrafish embryos, the authors compared the activity of TGFBR2a variants with just the inner or outer ECD, yet in the LBSA and reporter assays (Fig. 7C, 7D), these TGFBR2a variants were not examined. In light of this, the authors should repeat the LBSA and reporter assays shown in 7C and 7D so as to include the variants of TGFBR2a with either just inner or outer ECD.

2. In the discussion it is stated that an advantage of tandem ECDs is that this could enable more efficient capture of ligands when the ligand concentrations are low – this is on the one hand an intriguing possibility – on the other hand, one of the most glaring omissions throughout the manuscript is an articulation of the linkers connecting the ECDs together – this is highly relevant as this would determine whether the type of mechanism that the authors propose (where tandem ECDs are binding, for example, a single TGF- β homodimer) might be possible. In light of this, the authors should modify their manuscript to describe in more detail the linkers in each of the ECD duplicated constructs they describe – in addition they should comment, where relevant, if the linkers would enable binding of tandem ECDs to a single ligand homodimer (or not) – this is a very important point and may possibly be related to the different behavior (sensitizing or inhibitory) of the tandem ECD constructs from different species that the authors describe.

3. In a related vein, the authors should describe the linker in their synthetic TGFBR2 construct and whether this would permit binding of the tandem ECDs to a single ligand homodimer or not. If it does, what behavior would be observed if the linker is shortened so that it does not permit binding of the tandem ECDs to a single ligand? If it does not, what behavior would be observed if the linker is extended so that it does permit binding of the tandem ECDs to a single ligand?

4. In Fig. 3G, the signaling activity of hBMPR2, which is presumably the positive control, is practically unchanged from the negative control, which seems inconsistent.

5. It is stated that all key TGFBR2 interacting residues are conserved in the inner ECDs of all duplicated TGFBR2s – however in the *Xenopus* TGFBR2 Glu119 in the C-terminal segment, which is known to be a critical residue, is non-conservatively substituted with a leucine – this should be commented upon and explained.

Reviewer #2

(Remarks to the Author)

This is a very well written manuscript that identifies an important evolutionary mechanism that fine tunes TGF- β signaling. The manuscript flows logically and is easy to follow. This work will be of high significance to the field and may explain difference in tissue specific signaling. One of the strengths of the study is that the authors use a combination of genomic, bioinformatic, in silico, and experimental data to identify multiplication of ligand binding domains in varying species and then show the consequences for ligand binding and signal function. They use structural and imaging techniques to provide deeply mechanistic mechanisms for the consequences of multimerization of ligand binding domains in various species. They show the multimerization of LBD is independent of whole gene duplication and the multiple ligand binding domains fine tunes signaling with slightly different outcome in different species, in some cases, extra binding domain attenuating signaling. They use the zebrafish model, to show tissue specific differences in expression and function. The methodology is sound and it looks like enough detail is provided to reproduce the experiments. The new sequencing data has been submitted to an appropriate data base. Overall, the study represents a large body of work that spans sequencing to hypothesis generation to experimental testing. The results are of high novelty and significance to understanding tissue specific variations in signal transduction and fine tuning of signaling through evolutionary processes.

Reviewer #3

(Remarks to the Author)

The authors clearly demonstrate that ligand-binding domain duplications have evolved as a mechanism for modulating TGF β superfamily signaling, an interesting finding but perhaps one that is better suited for a speciality audience interested in evolutionary biology.

What is lacking for me is any data to validate the statement by the authors that "these natural strategies may inform the rational design of synthetic TGF β modulators, such as multivalent ligand traps, with potential applications in regenerative medicine and cancer therapy." At present there are many ligand traps in clinical trials and it is unclear how additional ones, designed using the information contained here, would provide a significant benefit over currently available agents. The authors might be encouraged to develop data that supports this idea using a clinically relevant in vivo model.

Version 1:

Reviewer comments:

Reviewer #1

(Remarks to the Author)

The authors are to be commended for responding in a positive way to the critiques that were raised -as a result the manuscript is strengthened and the conclusions are supported with greater clarity.

Dear Reviewers,

We sincerely thank you for your careful evaluation of our manuscript and for considering it for publication in Nature Communications. We have addressed all comments and suggestions raised by the three reviewers and believe that the revisions have significantly strengthened the manuscript.

We are grateful for the opportunity to perform additional simulations and experiments, which have further improved the robustness of our findings. Please find below our detailed, point-by-point responses to each of the reviewers' comments.

On behalf of all coauthors, we greatly appreciate your time and thoughtful feedback.

With kind regards,

Jerome Jatzlau and Petra Knaus

Response letter- color code

Reviewer question Q1

Comment of authors in blue

Response letter

REVIEWER COMMENTS

Reviewer #1 (Remarks to the Author):

The manuscript by Jatzlau and co-workers is focused on building upon the prior observation by the same group that in medaka (Japanese rice fish), a single exon coding for the TGF- β family type I receptor ACVR1 underwent triplication to produce a receptor with three tandem ligand binding ECDs.

Through analysis of genomic sequence information of related fish, and other species, such as frogs, horses, and chickens, the authors have identified 12 more instances of ECD multiplication in three TGF- β family receptors, the type I receptor ACVR1 (and a close paralog ACVR1L) and the type II receptors BMPR2 and TGFBR2 (and its paralog TGFBR2b), which for each ECD are encoded by two exons.

Through sequence analysis, modeling of the complexes of BMPR2 and TGFBR2 with their high affinity ligands, ActA and TGF- β 1, respectively, as well as cell surface fluorescence-mediated Ligand Binding Surface Assays (LBSA assay), and reporter assays, the authors generally find that the inner-most ECD is the one that is most highly conserved, in terms of its ligand binding residues, relative to the known human-ligand complex structure, and the one that has the greatest binding affinity and signaling activity – conversely, the more distal domains have lower conservation of interacting residues, lower surface affinities, and lower signaling activity. Interestingly though, there are some exceptions to this, for example in chickens, the outmost TGFBR2 ECD is essentially just as conserved as the innermost ECD and its found that although the tandem ECD confers high surface affinity, the signaling output is nonetheless comparable to that of a single ECD. In zebrafish, it was found that there are two TGFBR2 genes, which are differentially expressed, TGFBR2b, which has a single, but evidently higher affinity ECD, and TGFBR2a, which has tandem ECDs, with the outer ECD attenuating the ligand binding and signaling output. In developing zebrafish embryos, the more actively signaling TGFBR2b variant, or the TGFBR2a variant with the outer ECD deleted, were characterized by increased ventralizing activity, suggesting that the two TGFBR2 genes may have evolved to fine tune TGF β signaling to enable different roles in vivo.

The authors conclude that ECD duplication can enable neofunctionalization (and thus tuning of signaling), which is indeed an interesting observation that has not previously been appreciated for TGF- β family signaling.

Reviewer 1 Q1: Overall, while I think this is an important finding that merits publication, there are nonetheless a few major and several minor questions that I think should be addressed prior to publication – one of the major questions relates to the broader significance of these findings – that is if ECD duplication is enabling neofunctionalization, and presumably with this advantages to the organism, why is ECD duplication not been more fruitful in evolution (i.e. an adaptation that has become more widespread as organisms have diversified)? Another major question is why has this occurred in just these genes – why not also in other related type I and type II receptors of the TGF- β family?

We thank the reviewer for raising this important question regarding the evolutionary distribution of extracellular domain (ECD) duplications. In the revised manuscript, we have expanded the Discussion to address why ligand-binding domain (LBD) duplication appears recurrent but remains restricted to a subset of TGF β receptors.

Briefly, several factors likely contribute to this pattern. First, the architecture of TGF β receptor complexes imposes strong structural constraints on receptor extracellular domains, such that domain multiplication could disrupt ligand engagement or receptor–receptor interactions in many receptors. The receptors identified in our study (ACVR1, BMPR2, and TGFBR2) may therefore represent a subset whose extracellular organization tolerates LBD duplication while preserving signaling competence. In some cases, this tolerance may be further facilitated by the co-expression of functionally conserved paralogous partners (e.g., ACVR1L for ACVR1 or ACVR2A/B for BMPR2), which could buffer architectural changes. By contrast, TGFBR2 acts as the sole high-affinity type II receptor transmitting canonical TGF β signaling, suggesting different evolutionary constraints.

Second, we note that extracellular domain duplication is not unique to vertebrates, as similar events have been reported in ancestral TGFBR2 orthologs in invertebrates such as sponges and oysters, indicating that domain multiplication represents a recurrent evolutionary strategy rather than a vertebrate-specific innovation.

Finally, we also note that current genome assemblies and transcriptomic datasets remain uneven across vertebrate lineages, and structural variants affecting receptor extracellular domains may remain undetected in poorly sampled or fragmented genomes.

These points have now been incorporated as a new paragraph into the Discussion to clarify the broader evolutionary context of our findings.

“A remaining question is why LBD duplication has occurred repeatedly in only a subset of receptors while remaining rare across the broader TGF β receptor family. Intriguingly, domain multimerization is not limited to vertebrates. Independent extracellular domain duplications in ancestral TGFBR2 orthologs have been described in invertebrates such as sponges and oysters^{47,48}, indicating that LBD duplication is an evolutionarily widespread strategy, independent of WGDs. However, within vertebrates, sporadic occurrence of LBD duplication in only a subset of receptors, suggests strong structural constraints on evolvability, with the affected receptors uniquely tolerant of domain multiplication while preserving initial ligand binding/signaling integrity in certain organisms. Tolerance of LBD duplication in some receptors may be facilitated by the co-expression of functionally conserved paralogous partners, such as ACVR1L alongside ACVR1 or ACVR2A/B alongside BMPR2. In contrast, TGFBR2 functions as the sole high-affinity type II receptor transmitting canonical TGF β signaling. At the same time, the apparent restriction of LBD duplications to these receptors should be interpreted cautiously. Current genome assemblies and transcriptomic datasets remain uneven across the vertebrate phylogeny, and structural variants affecting ECDs may remain undetected in poorly

sampled or fragmented genomes. Thus, additional examples of receptor domain multiplication may exist but fall below the current limits of detection. “

Other more minor (but nonetheless important to address) questions are as follows:

Reviewer 1 Q2: In the studies comparing zebrafish TGFBR2b and TGFBR2a in zebrafish embryos, the authors compared the activity of TGFBR2a variants with just the inner or outer ECD, yet in the LBSA and reporter assays (Fig. 7C, 7D), these TGFBR2a variants were not examined. In light of this, the authors should repeat the LBSA and reporter assays shown in 7C and 7D so as to include the variants of TGFBR2a with either just inner or outer ECD.

We thank the reviewer for raising this point and appreciate the opportunity to clarify this aspect of the study. The experiments including the TGFBR2a variants containing either the inner or the outer ligand-binding domain were indeed performed together with the LBD mutants of TGFBR2a and TGFBR2b. These data are presented in Fig. 5E and Supplementary Fig. 12 for the ligand surface binding assay (LSBA), and in Fig. 5F for the luciferase reporter assay. We have clarified this in the manuscript as shown below.

“Overall, the elevated ventralization observed *in vivo* is consistent with the higher signaling output measured *in vitro* for dTgfr2a LBDⁱⁿ and dTgfr2b (Fig. 5e,f; Supplementary Fig. 12; Fig. 7c,d), suggesting that increased dTgfr2 signaling capability contributes to the ventralizing effect (Fig. 7k).”

Figure 5: TGFBR2 LBD duplications differently finetune signaling competence. (E) hTGFβ1-SiR-d12 surface binding of *D. rerio* receptor variants represented as relative fluorescence intensity per area. (F) pSMAD2/3 sensitive CAGA₁₂ Luciferase reporter activity of *D. rerio* receptor variants as FI ± SD of RLU. Statistical significance was calculated using two-way ANOVA and (F) Tukey’s post-hoc test within groups or (E) Dunnett post-hoc test relative to control (**p < 0.01, ***p < 0.001, ****p < 0.0001).

Supplementary Figure 12: Representative LSBA images of TGFβ1 bound to dTgfb2a/b variants. Representative confocal microscopy images of LSBA for unstimulated and hTGFβ1-SiR-d12 stimulated COS-7 cells expressing human BMP2 or *danio rerio* variants (dTgfb2b, dTgfb2a (2LDB), dTgfb2a-LDBⁱⁿ, dTgfb2a-LDB^{out}) or untransfected control. Scale bar \approx 20 μ m.

Reviewer 1 Q3: In the discussion it is stated that an advantage of tandem ECDs is that this could enable more efficient capture of ligands when the ligand concentrations are low – this is on the one hand an intriguing possibility – on the other hand, one of the most glaring omissions throughout the manuscript is an articulation of the linkers connecting the ECDs together – this is highly relevant as this would determine whether the type of mechanism that the authors propose (where tandem ECDs are binding, for example, a single TGF- β homodimer) might be possible. In light of this, the authors should modify their manuscript to describe in more detail the linkers in each of the ECD duplicated constructs they describe – in addition they should comment, where relevant, if the linkers would enable binding of tandem ECDs to a single ligand homodimer (or not) – this is a very important point and may possibly be related to the different behavior (sensitizing or inhibitory) of the tandem ECD constructs from different species that the authors describe.

We are very grateful to the reviewer for this insightful and important comment. We fully agree that linker architecture is a critical determinant of the binding mode of tandem ECD constructs and appreciate the opportunity to clarify this aspect of our study.

In response, we have substantially expanded the manuscript to explicitly describe the linker regions connecting duplicated LBDs across all analyzed TGFBR2 and BMRP2 variants (**NEW Supplementary Fig17c, NEW Fig.6c**). In addition, we performed structural modeling to assess whether linker length permits intra-ligand bivalency (i.e., simultaneous engagement of both receptor-binding sites within a single TGF β homodimer). These analyses indicate that a minimal linker length of ~36 residues is required to enable such a configuration (**NEW Fig.6b**).

Importantly, comparison with naturally occurring linker lengths across species reveals that most variants possess substantially shorter linkers, rendering intra-ligand bivalency unlikely and favoring inter-ligand binding (**NEW Fig.6c**). Notably, even in cases with longer linkers (e.g., *C. harengus*), sequence divergence within the membrane-distal LBD suggests reduced ligand-binding capacity, further limiting the feasibility of intra-ligand bivalency.

These additions are now incorporated into the Results section and discussed accordingly as highlighted below.

“After characterizing the differential ligand-binding capacity of LBD-duplicated TGFBR2 variants, we sought to predict the impact of linker length between the two LBDs. While two identical LBDs may enable inter-ligand binding, thereby increasing total ligand engagement, a flexible linker of sufficient length could permit intra-ligand bivalency. In this scenario, two LBDs within a single receptor would occupy both type II receptor-binding sites, potentially interfering with the formation of a functional tetrameric receptor complex (Fig. 6a, Supplementary Fig. 17a-b). To estimate the linker length required for intra-ligand bivalency, we modeled human TGFBR2 variants containing two identical LBDs connected by linkers of increasing length, starting from the 20-residue linker encoded at the E3–E2' junction. Linkers of ≥ 36 residues were predicted to represent the minimal requirement for intra-ligand bivalency (Fig. 6b). Comparison with naturally occurring linker lengths across TGFBR2 variants revealed that most species possess linkers shorter than 24 residues, with the exception of *C. harengus* (44 residues) (Fig. 6c, Supplementary Fig. 17c). While short linkers may permit inter-ligand binding, intra-ligand bivalency is therefore unlikely in most species. Notably, the membrane-distal LBD in *C. harengus* lacks key residues of the TGF β -binding interface, consistent with reduced binding affinity (Supplementary Fig. 7,11a), further arguing against intra-ligand bivalency. “

PART of Figure 6: Human TGFBR2 LBD duplication enhances ligand binding without affecting downstream signaling. (a) Schematic illustration of two potential modes of ligand binding by 2LBD receptors: inter-ligand binding and intra-ligand bivalency. Intra-ligand bivalency may require a sufficiently long and flexible linker between the two LBDs. Linker length is defined by the residues encoded at the junction of exons E3 and E2'. (b) Structural predictions assessing the minimal linker length required for intra-ligand bivalency using hTGFβ1 and a modeled hTGFBR2 2LBD. The number of linker residues embedded within the dimer interface is plotted against total linker length for the top three ranked models. Consistent across these predictions, a linker length of 36 residues did not result in threading through the dimer interior and was therefore defined as the predicted cutoff for intra-ligand bivalency. (c) Comparison of naturally occurring linker lengths between 2 LBDs across TGFBR2 variants from different species. Most variants fall below the predicted cutoff for intra-ligand bivalency, suggesting that intra-ligand bivalency is not feasible and that TGFβ1 binding could occur via an inter-ligand mode.

Supplementary Figure 17: Linker architecture in TGFBR2 and BMPR2 variants. (a) Schematic illustration of exon architecture of receptors with duplicated (i.e. TGFBR2) or triplicated (i.e. BMPR2) extracellular LBDs. Linker length is defined by the residues encoded at the junction of exons E3 and E2'. (b) Illustration of two potential modes of intra-ligand bivalency of receptors with two or three LBDs. Intra-ligand bivalency may require a sufficiently long and flexible linker between the LBDs. The presence of a third LBD could further affect the likelihood of bivalent binding. (c) Cross-species overview of linker length variation in duplicated LBD TGFBR2 and BMPR2 variants.

Reviewer 1 Q4: In a related vein, the authors should describe the linker in their synthetic TGFBR2 construct and whether this would permit binding of the tandem ECDs to a single ligand homodimer or not. If it does, what behavior would be observed if the linker is shortened so that it does not permit binding of the tandem ECDs to a single ligand? If it does not, what behavior would be observed if the linker is extended so that it does permit binding of the tandem ECDs to a single ligand?

We thank the reviewer for this thoughtful and constructive comment, which raises an important point regarding the functional implications of linker architecture in our synthetic construct.

In the revised manuscript, we now explicitly describe the linker present in the engineered human TGFBR2 2LBD construct (**NEW Supplementary Fig17c, NEW Fig.6c**). This linker corresponds to the native sequence generated by E2–E3 duplication and is shorter than the ~36-residue threshold predicted to permit intra-ligand bivalency. Consistent with this, our structural modeling indicates that this construct is unlikely to support simultaneous engagement of both LBDs with a single TGF β homodimer and instead favors inter-ligand binding (**NEW Fig.6a-b**).

Experimentally, this is reflected in enhanced ligand binding but attenuated downstream signaling, as evidenced by reduced short-term SMAD2 activation despite preserved transcriptional output at later time points (**Fig. 6d–h**). These findings support a model in which increased ligand capture via inter-ligand binding limits productive receptor complex formation.

Our structural modeling suggests that extending the linker beyond the predicted threshold could permit intra-ligand bivalency, allowing both LBDs within a single receptor to engage a single TGF β homodimer. Such a configuration would be expected to further reduce signaling output by sequestering ligand in a non-productive complex that limits efficient receptor assembly. Importantly, inter-ligand binding would not be precluded under these conditions and may still occur in parallel. Notably, our comparative analysis indicates that naturally occurring linker lengths in TGFBR2 variants rarely reach this threshold, suggesting that this binding mode is unlikely to be prevalent *in vivo*. Moreover, intra-ligand bivalency would require both LBDs to retain comparable ligand-binding capacity, whereas in most naturally occurring variants ligand binding is constrained by the membrane-distal domain, further limiting the feasibility of this mechanism.

This aspect has been incorporated into the revised Results section.

“Next, we engineered a synthetic human TGFBR2 variant with two identical LBDs which would result from E2-E3 duplication and assessed its ability to bind human TGF β 1. Consistent with the chicken receptor, LBD duplication significantly enhanced ligand binding, confirming the inter-ligand binding mode (Fig. 6d-e, Supplementary Fig. 18). Whereas CAGA₁₂-luc activity after 24 hours was comparable for both TGFBR2 variants (Fig. 6f), short term activation of SMAD2 was reduced in the presence of TGFBR2 2 LBD compared to TGFBR2 expression (Fig. 6g). Collectively, this highlights that 2 LBD TGFBR2 receptors remain functional but limit the downstream SMAD activation through excessive ligand binding (Fig. 6h)”.

Reviewer 1 Q5: In Fig. 3G, the signaling activity of hBMPR2, which is presumably the positive control, is practically unchanged from the negative control, which seems inconsistent.

We agree with the reviewer that, at first glance, the relationship between ligand binding and signaling output may appear inconsistent, and we therefore sought to clarify this aspect experimentally.

As stated in the text: “ Whereas BMP binding to BMPR2-containing receptor complexes requires the presence of a high affinity type I receptor, BMPR2 can directly bind Activin A through a specific set of hydrogen bonds and hydrophobic interactions primarily involving residues located between fingers 2 and 3 (Fig. 3b, c)¹⁹.”

However, although Activin A can bind BMPR2 independently of a type I receptor, this does not necessarily mean that it can activate a specific SMAD signaling branch downstream.

In order to clarify this discrepancy for human BMPR2, we performed additional CAGA₁₂-luciferase reporter assays to assess SMAD2/3 signaling in the presence of different type II receptors with varying ligand-binding properties (Rebuttal Fig. 1). We find that human BMPR2 does not enhance SMAD2/3 signaling, consistent with its limited affinity for Activin A compared to other type II receptors such as ACVR2A and ACVR2B (Jatzlau et al. 2023, DOI: 10.1038/s42003-022-04388-4). In contrast, the high-affinity Activin receptor ACVR2B modestly enhances signaling at low ligand concentrations. However, this effect is not sustained at higher ligand concentrations, where signaling from endogenous receptors saturates the SMAD2/3 response. Notably, BMPR2 expression results in a reduction of SMAD2/3 signaling under these conditions.

Rebuttal Fig. 1: pSMAD2/3 sensitive CAGA₁₂ Luciferase reporter activity of Halo-hBMPR2 and hACVR2B as FI ± SD of relative luminescence units (RLU). Statistical significance was calculated using two-way ANOVA and Dunnett post-hoc test relative to control (**p < 0.01, ****p < 0.0001).

These observations are consistent with differences in receptor complex formation, as ACVR2 receptors are known to efficiently partner with ALK4 (Olsen et. al 2015, DOI: 10.1186/s12964-015-0104-z), whereas BMPR2 is less likely to form productive signaling complexes in the Activin context, potentially leading to ligand sequestration and reduced signaling. In addition, BMPR2 has been reported to form non-signaling complexes with the type I receptor ALK2, which may further contribute to the observed repression of SMAD2/3 signaling (Olsen et. al 2018, DOI: 10.1242/jcs.213512).

Interestingly, the BMPR2 ortholog from elephant fish enhances both Activin binding and SMAD2/3 signaling, resembling the behavior of ACVR2-type receptors.

We clarified this in the results text as follows: “Interestingly, the BMPR2 ortholog from elephant fish enhanced both Activin A binding and SMAD2/3 signaling, whereas the human BMPR2 control increased ligand binding but failed to efficiently promote SMAD2/3 activation. This functional divergence suggests that the elephant fish BMPR2 exhibits ACVR2-like receptor behavior²².”

Reviewer 1 Q6: It is stated that all key TGFBR2 interacting residues are conserved in the inner ECDs of all duplicated TGFBR2s – however in the *Xenopus* TGFBR2 Glu119 in the C-terminal segment, which is known to be a critical residue, is non-conservatively substituted with a leucine – this should be commented upon and explained.

We thank the reviewer for pointing out the unique feature of the *Xenopus* LBDⁱⁿ. As shown in Supplementary Fig.9d-f the *Xenopus* LBDⁱⁿ facilitates TGFβ1 binding and can enhance signaling. However, the downstream CAGA12 luc response is slightly less compared to the human TGFBR2 receptor which could reflect the difference at position 119 (Supp. Fig.9F).

We addressed the point raised by the reviewer in the results section as shown below.

“Similarly, in horse and frog TGFBR2 orthologs, the membrane-proximal domain LBDⁱⁿ shows the highest predicted and experimentally confirmed binding to TGFβ1, while the outer domain (LBD^{out}) exhibits reduced or no binding and lacks signaling competence (Fig. 5g-i, Supplementary Fig. 9d-f, 14, 15). Notably, although most key TGFBR2 human TGFβ1-contact residues are conserved within the LBDⁱⁿ domains, the frog LBDⁱⁿ contains a non-conservative substitution at position 119 (E119L) within the C-terminal segment previously implicated in ligand interaction (Fig. 4c)²³. Despite this substitution, the LBDⁱⁿ domain facilitates ligand binding, suggesting that the overall binding interface remains functionally preserved, potentially through compensatory interactions or structural tolerance within the receptor–ligand complex. Importantly, the presence of the LBD^{out} does not impair binding or signaling function of the full length TGFBR2 for either species (Fig. 5h-i, Supplementary Fig. 9e-f, 14, 15).

”

Supplementary Figure 9: Functional implications of LBD multiplication in TGFBR2 orthologs of *Xenopus laevis*. (D) *in silico* binding analysis via Rosetta docking of hTGFβ1 to inner (pink) and outer LBD (blue) of *X. laevis* depicted as interface score (REU) in relation to interface root mean square deviation (iRMSD) in angstroms. REU differences of in and out are calculated by respective mean REU values of each LBD. (E) hTGFβ1-SiR-d12 surface binding of *X. laevis* receptor variants represented as relative fluorescence intensity per area. (F) pSmad2/3 sensitive CAGA₁₂ Luciferase reporter activity of *X. laevis* receptor variants as FI ± SD of relative Luminescence Units (RLU). Statistical significance was calculated using two-way ANOVA and (E) Dunnett post-hoc test relative to xITgfr2.L

(E) or using two-way ANOVA and (F) Tukey's post-hoc test within groups (*p < 0.05, **p < 0.01, ***p < 0.001, ****p < 0.0001).

Reviewer #2 (Remarks to the Author):

This is a very well written manuscript that identifies an important evolutionary mechanism that fine tunes TGF- β signaling. The manuscript flows logically and is easy to follow. This work will be of high significance to the field and may explain difference in tissue specific signaling. One of the strengths of the study is that the authors use a combination of genomic, bioinformatic, in silico, and experimental data to identify multiplication of ligand binding domains in varying species and then show the consequences for ligand binding and signal function. They use structural and imaging techniques to provide deeply mechanistic mechanisms for the consequences of multimerization of ligand binding domains in various species. They show the multimerization of LBD is independent of whole gene duplication and the multiple ligand binding domains fine tune signaling with slightly different outcome in different species, in some cases, extra binding domain attenuating signaling. They use the zebrafish model, to show tissue specific differences in expression and function. The methodology is sound and it looks like enough detail is provided to reproduce the experiments. The new sequencing data has been submitted to an appropriate data base. Overall, the study represents a large body of work that spans sequencing to hypothesis generation to experimental testing. The results are of high novelty and significance to understanding tissue specific variations in signal transduction and fine tuning of signaling through evolutionary processes.

We thank the reviewer for this thoughtful assessment of our work. We greatly appreciate the recognition of the conceptual advance, the integrative nature of our approach, and the effort to combine genomic, computational, structural, and experimental analyses to address this question. We are particularly encouraged that the reviewer highlights the potential relevance of our findings for understanding tissue-specific modulation of TGF β signaling, as well as the evolutionary perspective of domain-level innovation. We also appreciate the positive comments on the clarity of presentation, methodological rigor, and data availability.

Overall, we are grateful for the reviewer's careful evaluation and supportive feedback.

Reviewer #3 (Remarks to the Author):

The authors clearly demonstrate that ligand-binding domain duplications have evolved as a mechanism for modulating TGF β superfamily signaling, an interesting finding but perhaps one that is better suited for a speciality audience interested in evolutionary biology.

What is lacking for me is any data to validate the statement by the authors that "these natural strategies may inform the rational design of synthetic TGF β modulators, such as multivalent ligand traps, with potential applications in regenerative medicine and cancer therapy." At present there are many ligand traps in clinical trials and it is unclear how additional ones, designed using the information contained here, would provide a significant benefit over currently available agents. The authors might be encouraged to develop data that supports this idea using a clinically relevant in vivo model.

We thank the reviewer for this thoughtful and important comment regarding the translational implications of our findings. We agree that our original statement was speculative and not

directly supported by experimental data within the scope of this study. In response, we have revised the final paragraph of the Discussion to remove references to potential clinical applications and instead focus on the conceptual advance provided by our findings. The revised text now emphasizes LBD multimerization as a general mechanism to tune ligand engagement and signaling specificity within the TGF β pathway. We believe this revision more accurately reflects the scope of our data and avoids overinterpretation.

The changes are now reflected in the revised discussion as shown below:

“Together, these findings reveal that LBD duplication acts as a versatile mechanism for modulating TGF β -receptor function, enabling the evolution of context-sensitive signaling outputs. Given the pivotal role of TGF β -signaling in regulating cell fate, plasticity, and migration, such domain-level adaptations likely contributed to the evolutionary diversification of vertebrate body plans. More broadly, these results establish LBD multimerization as a general mechanism to tune ligand engagement and signaling specificity within the TGF β pathway.”